

# The fossil bivalve *Angulus benedeni benedeni*: a potential seasonally resolved stable isotope-based climate archive to investigate Pliocene temperatures in the southern North Sea basin

Nina M.A. Wichern[1], Niels J. de Winter[2,3], Andy L.A. Johnson[4], Stijn Goolaerts[5], Frank Wesselingh[2,6],
Maartje F. Hamers[2], Pim Kaskes[3], Philippe Claeys[3], Martin Ziegler[2]

[1]Institute of Geology and Paleontology, Westfälische Wilhelms-Universität, Münster, 48149, Germany
[2]Department of Earth Sciences, Utrecht University, Utrecht, 3584 CB, the Netherlands
[3]Analytical, Environmental, and Geochemistry Research Group, Department of Chemistry, Vrije Universiteit Brussel, Brussels, 1050, Belgium
[4]School of Environmental Sciences, University of Derby, Derby, DE22 1GB, UK
[5]Department of Palaeontology, Royal Belgian Institute of Natural Sciences, Brussels, 1000, Belgium
[6]Naturalis Biodiversity Center, Leiden, 2333 CR, the Netherlands

Correspondence to: Nina M.A. Wichern (nwichern@uni-muenster.de)



**Abstract.** Obtaining temperature data from the mid-Piacenzian warm period (ca. 3 Ma, Pliocene epoch) is a key factor in outlining the impact of projected anthropogenic climate change. The mid-Piacenzian warm period was a high-$CO_2$ world with a paleogeography similar to modern times. The time interval has been used to validate and improve climate model retrodictions, which in turn enables assessing the predictive strength of these models. Validating climate models requires a

large array of robust proxy data. Here, we increase the potential of this proxy database by showing that the extinct tellinid bivalve *Angulus benedeni benedeni* can be used for stable isotope-based temperature reconstructions. This species is found in the Pliocene sediments of the southern North Sea basin. Oxygen isotope and carbonate clumped isotope measurements on the shell of *A. benedeni benedeni* resulted in a mean annual temperature reconstruction of 13.5±3.8°C. This is 2.5°C warmer than today, 3.5°C warmer than the pre-industrial North Sea, and in line with global Pliocene temperature estimates of +2-4°C

compared to the pre-industrial climate. Limited amounts of clumped isotope data hindered determining summer and winter temperatures, but the oxygen isotope record shows that the growth band spacing of *A. benedeni benedeni* allows for sampling at a resolution of 2-3 months. The species could live for up to a decade, and therefore has the potential to be used for multiannual seasonality reconstructions. The pristine nature of the aragonitic shell material was verified through electron backscatter diffraction analysis (EBSD), and backed by light microscopy, X-ray diffraction, and X-ray fluorescence. The

various microstructures as obtained from the EBSD maps have been described, and they provide a template of pristine *A. benedeni benedeni* material to which potentially altered shells may be compared. The bivalve *A. benedeni benedeni* is suitable for high resolution isotope-based paleoclimatic reconstruction and it can be used to unravel the marine conditions in the Pliocene southern North Sea basin at a seasonal scale, yielding enhanced insights into imminent western European climate conditions.





## 1 Introduction

During the Pliocene-Pleistocene transition, global climate changed from a "coolhouse" state to the icehouse state (Westerhold et al. 2020). One specific period in the late Pliocene, the mid-Piacenzian warm period (mPWP—3.264-3.025 Ma; Dowsett et al. 2010), is a period of interest in light of our current warming climate. During the mPWP, global average temperatures were 2-4°C warmer than today (Ravelo et al. 2004; Haywood and Valdes 2004; Dowsett et al. 2010; 2012; Haywood et al. 2013; 2020), atmospheric $CO_2$ concentrations were 340-380 ppm (360 ppm on average—Raymo et al. 1996; Kürschner et al. 1996; Pagani et al. 2010; de la Vega et al. 2020), and sea level was 10-30 m higher (Ravelo et al. 2004; Naish and Wilson 2009; Dowsett et al. 2010). The cause of warming during this period is still debated. Increased $CO_2$ levels and ocean circulation changes, or a combination thereof, are the dominant hypotheses (e.g., Raymo et al. 1996; Mudelsee and Raymo 2005; Dowsett et al. 2009; De Schepper et al. 2014). These oceanic circulation changes were brought on by the closing, restriction, or through-flow change of several seaways (respectively, the Isthmus of Panama, the Indonesian Seaway, and the Bering Strait), resulting in the redirection of heat flows (e.g., Mudelsee and Raymo 2005; De Schepper et al. 2014; Horikawa et al. 2015). The estimated warming during the mPWP is similar to what has been predicted for the end of this century according to the moderate and high emission RCP scenarios (RCP4.5: average of 2°C warming, RCP6.0: average of 2.5°C warming, RCP 8.5: average of 4°C warming; Collins et al. 2013). Direct comparisons between the mPWP and our future climate are not appropriate, however. The mPWP was an equilibrated rather than a transient climate (e.g., Salzmann et al. 2009), and some palaeogeographical features—such as seaways configurations—that have changed since could have had a significant impact on regional climate (Hill 2015).

Validating climate models requires large, high-quality datasets. An example is the PRISM project (Dowsett et al. 2010; 2016), which is a compilation of data on paleogeography, land and sea temperatures, vegetation cover, land and sea ice, and soil type for the mPWP. Assessing the suitability of bivalve species that have not yet been used for this purpose is a way to potentially increase this database. To this goal, we investigated specimens of the extinct bivalve species *Angulus benedeni benedeni* (Nyst and Westendorp 1839) that were collected from the mid-Piacenzian sediments of the southern North Sea basin. Investigating its potential use as a climate archive was done by means of oxygen isotope and carbonate clumped isotope analyses. In addition, electron backscatter diffraction in combination with light microscopy, X-ray diffraction, and micro-X-ray fluorescence, was used to analyse microstructures and assess preservation. The results indicate that *A. benedeni benedeni*'s shell contain multiannual records. Sampling these at a sub-annual resolution allows for the reconstruction of the average temperature and seasonality of its shallow marine habitat, showing that *A. benedeni benedeni* can be a valuable addition to the mid-Piacenzian climate archive of western Europe.





## 2 Background

### 2.1 Seasonality records from bivalve shells

Reconstructions of past high-CO₂ worlds deliver crucial insights into the processes, rates, and outcomes that are associated with anthropogenic climate change. The knowledge we have of these past climate modes, however, is often limited to long-term climate records such as deep-sea benthic oxygen isotope curves (e.g., Westerhold et al. 2020). This long-term record bias limits our understanding of climate impact on shallow marine and terrestrial ecosystems. This is especially relevant as

short-term events such as heatwaves and extreme precipitation have already been observed to have increased in frequency and are predicted to become even more common in the future (Seneviratne et al. in Press). Under a warming climate, seasonality is expected to intensify as well. Climate models predict an enhanced seasonal contrast in both temperature and precipitation (Seneviratne et al. in Press). Moreover, seasonality encompasses the most important periodicity in climate and is therefore crucial to our understanding of past and future climates (Mitchell 1976; Huybers and Curry 2006; von der Heydt

et al. 2021). Many proxies provide insight into mean annual temperatures, but these data do not capture the range of temperatures that a region experiences. Ontogenetic profiles from bivalves have the potential to record climate variability at the sub-annual scale. Bivalve shells can be suitable archives for seasonality reconstructions for several reasons. Long-lived species record multiple seasons as they live for several years—up to 500 years in certain species (Butler et al. 2013)—and can record down to sub-daily environmental conditions in their growth increments (e.g., Schöne et al. 2002). They are

abundant in the fossil record across all latitudes (Moss et al. 2016 and references therein). Through sclerochronologic analysis of bivalve shells, the growth patterns can be used to infer life history (e.g., Sato 1999; Palmer et al. 2021). Many bivalves occupy the shallow marine zone. This zone is less well-represented than the open ocean in climate records (e.g., de Winter et al. 2020 and references therein). Shallow marine and coastal records can also be used to connect terrestrial and marine records, as they collect input from both realms (Crampton-Flood et al. 2020). Most bivalve species precipitate

isotopically in near equilibrium with water and are relatively little affected by so-called "vital effects" that skew the temperature calibration (Weiner and Dove 2003; Huyghe et al. 2022).

### 2.2 The southern North Sea basin

The southern North Sea basin (SNSB) has already warmed by 1-1.3°C since the late 1800s (from a mean annual average temperature of ca.10°C to 11°C), with the majority of the warming having taken place since the late 1980s (Schöne et al.

2004; Mackenzie and Schiedek 2007; Belkin 2009; Emeis et al. 2015; Quante and Colijn 2016). Within the North Sea the rate of warming varies regionally, ranging from 0.2°C/decade in the north to up to 0.5°C/decade in the south since the 1980s (Quante and Colijn 2016). North Sea ecosystems are already affected by these increasing temperatures (Quante and Colijn 2016).

Besides its response to recent climate change, it is also an important region to understand in the context of past climate

change through its connection to the North Atlantic. Proxy data indicate strong warming in the North Atlantic region during





the mid-Piacenzian, with global mean sea-surface temperature (SST) anomalies of up to +4-7°C instead of the global +2-4°C (Dowsett et al. 2012; 2013). Modelling studies have often failed to reproduce this enhanced sensitivity (Dowsett et al. 2013), although recent studies have obtained a better reconciliation of proxy and model data (Haywood et al. 2020). The North Atlantic is sensitive to changes in the Atlantic Meridional Overturning Circulation (AMOC) (Chen and Tung 2018). The

Gulf Stream, a component of the AMOC, extends north-eastward as the North Atlantic Current (NAC). This NAC water, which retains the warm and nutrient-rich signature of the Gulf Stream, enters the North Sea via the northern side and via the English Channel (Winther and Johannessen 2006). During the mPWP, only the northern route existed, as modern-day France and England were connected by the Weald-Arrtois land bridge to the south (e.g., Gibbard and Lewin 2016). Understanding the mid-Piacenzian North Sea climate could provide indirect evidence on North Atlantic circulation modes. Insights in

seasonality in particular could help us understand the distribution of heat throughout the year and how that relates to oceanic and atmospheric circulation patterns.

However, previous research into the SNSB climate and seasonality during the Pliocene, based on a range of biogenic proxies, has produced conflicting results. Some studies indicate generally cool winter conditions, with only slightly higher summer temperatures and a slightly larger seasonal range compared to today (Vignols et al. 2019). Other studies suggest

cool winter conditions similar to today with significant warming during summer, yielding significantly larger seasonal temperature ranges of up to 17°C (Raffi et al. 1985; Johnson et al. 2009; Valentine et al. 2011; Johnson et al. 2022). Yet another range of studies suggest strong warming in both summer and winter (Strauch 1968; Head 1997; 1998). Finally, some suggest a decreased seasonality compared to today (Wood et al. 1993; O'Dea and Okamura 2000; Knowles et al. 2009). Clumped isotope data from bivalves could constrain seasonal temperature range, as it enables a high-resolution, absolute

temperature reconstruction that does not rely on estimates of oceanic isotopic composition (see also sect. 3.5.3).

## 2.3 Geological context: the Lillo Formation

Three specimens of *Angulus benedeni benedeni* (SG-125, SG-126, and SG-127), were collected in 2013 from a shell bed in the top of Oorderen Member of the Pliocene Lillo Formation. The exact location was the temporary outcrop for the construction of the Deurganck Dock Lock (now Kieldrecht Lock; 51°16′44″N 4°14′52″E) in the port of Antwerp area,

northern Belgium (Fig. 1a, b). Photos of the collecting locality with an in-situ specimen of *A. benedeni benedeni* can be found in Deckers et al. (2020)'s Fig. 6. The marine Lillo Formation (De Meuter and Laga 1976) was deposited during the late Zanclean to the Piacenzian (ca. 3.7-2.7 Ma) in the SNSB (Fig. 1c). The Oorderen Member as introduced by De Meuter



and Laga (1976) is a greyish fine-grained shell-rich unit containing numerous mollusc shells, both dispersed in a

Figure 1: Location of the modern North Sea basin and the stratigraphic level from which the *A. benedeni benedeni* specimens originated. (a) Overview map of the location of Belgium and the present-day extent of the North Sea within northwest Europe. The dark rectangle indicates the location of map (b). Adapted after Louwye et al. (2020). (b) The geographical extent of the Lillo Formation in Belgium, after Louwye et al. (2020), and the location of the city of Antwerp. (c) Lithological column of the Lillo Formation and its members with their chronostratigraphic position, after Louwye et al. (2020). The approximate stratigraphic position of the *A. benedeni benedeni* specimens is shaded in grey.

glauconiferous quartz sand matrix as well as arranged into a number of cm to dm thick shell beds. Three intervals

characterised by different sedimentary structures (respectively: throughs and storm beds, predominantly homogenised sand,

predominantly bioturbated clayey sand) and mollusc fauna composition (respectively frequent occurrence of: *Atrina fragilis*



*kalloensis*, *Cultellus cultellatus*, *Angulus benedeni benedeni*), and separated by shell beds that experienced load coasting, can
be recognised all over the Port of Antwerp area, namely the *Atrina* level, the *Cultellus* interval and the *Angulus* interval (see
e.g., Vervoenen, Herman, and Van Waes 1995; Marquet and Herman 2009). The specimens have been collected from the
upper bioturbated *Angulus* clayey sand interval. The Oorderen Member has been interpreted to represent a neritic
environment (Louwye et al. 2004; De Schepper et al. 2009). Marquet (2004) estimated a water depth of 35-45 m based on
bivalve depth ranges, but the sedimentology of the Oorderen suggests a shallower depth. The member contains storm beds,
cross-stratification, tidal structures, and evidence of periods of emergence (Louwye et al. 2004; Louwye et al. 2020). We
therefore estimate a paleo water depth of 20 m. The increasing amount of clay towards the top of the Oorderen has been
interpreted to reflect a transition from calm water to a more high-energy tidal environment with clayey tidal lag deposits
(Louwye et al. 2004). Dinoflagellate assemblages point to a warming trend within the Oorderen Member, with warm-
temperate conditions in the upper part from which the specimens were collected (Louwye et al. 2004; De Schepper et al.
145    2009).

Biostratigraphically, the Oorderen Member lies within planktonic foraminifera biozone NPF16, benthic foraminifera biozone
B12, otolith zone 19, and benthic mollusc BM22B biozone (Vandenberghe and Louwye 2020). The Oorderen Member has
previously been interpreted as having been deposited during the mPWP (Louwye and De Schepper 2010; Valentine et al.
2011), and Louwye et al. (2020) place the Oorderen at ca. 3.71-3.15 Ma. This places its top, from which the specimens were
collected, within the 3.264-3.025 Ma mid Piacenzian Warm Period (mPWP) as defined by Dowsett et al. (2010).

### 2.4 *Angulus benedeni benedeni*

*Angulus benedeni benedeni* (Nyst and Westendorp 1839) (family Tellinidae) is one of the characteristic molluscs of the
Oorderen Member of the Pliocene Lillo Formation in Belgium, with an acme in the clayey upper part of the member
(Marquet 2005; Marquet and Herman 2009). It is the youngest member of an extinct North Sea basin lineage starting in the
late Eocene, with this subspecies emerging in the Pliocene (Marquet 2005; Moerdijk et al. 2010). It has been found in
Belgium, the Netherlands, and the southeastern United Kingdom (Marquet 2005). *Angulus benedeni benedeni* has not been
used in palaeoclimatological research before. Therefore, its potential as a climate archive is unknown, as is its life history:
the maximum age it could reach, its growth rate and how this rate changed throughout its life, the amount of time
represented in one growth increment, and possible periods of growth cessation.

### 3 Methods

#### 3.1 Light microscopy

Light microscopy was employed on all specimens to visually check for potential diagenetic alteration, to analyse the
macrostructures of *A. benedeni benedeni*'s shell, and to count growth increments. Two specimens were partially or fully
embedded in Araldite 2020 epoxy and high-polish thick (approximately 5 mm) sections were made, using a final polish of



0.3 µm aluminium oxide. These thick sections were viewed under a KEYENCE VHX-5000 digital microscope at x250 and x1000 magnification. Light microscopy composite images were then made with the Fiji stitching ImageJ plugin (Preibisch et al. 2009). These composites were used to count the growth increments.

### 3.2 X-ray diffraction

To reconstruct accurate growth temperatures through clumped isotope analysis, we had to verify that the original aragonitic
mineralogy was preserved. To check for diagenetic remineralization of original aragonite into calcite, powder X-ray diffraction (XRD) analysis of a representative section of one *A. benedeni benedeni* specimen (SG-125) was carried out on a Bruker D8 Advance diffractometer at Utrecht University that had been calibrated with a corundum crystal. The sample was ground to < 10µm in a Retsch McCrone mill with zirconium oxide grinding elements prior to front-loading it into a PMMA sample holder (diameter: 25 mm; depth: 1 mm). The instrument settings are given in table 1. The *A. benedeni benedeni*
pattern was compared to XRD spectra of pure aragonite and calcite samples. The calcite spectrum was recovered from the RRUFF database (Lafuente et al. 2015; RRUFF ID: R040070) and consisted of a powdered sample of a single calcite mineral from Pryor Mountain, Big Horn Country, Montana (USA) measured for X-Ray Diffraction at the University of Arizona Mineral Museum. For the pure aragonite spectrum, 1.12 g of shell from a common cockle (*Cerastoderma edule*) collected at the North Sea coast near Hoek van Holland (51°59'14" N, 4°05'56" E) was prepared and measured under the
same conditions as the fossil samples. XRD spectra of specimen SG-125 as well as the aragonite spectrum are available in the supplementary materials (see data availability).

| Instrument setting | Value |
|---|---|
| Voltage [kV] | 40 |
| Ampere [mA] | 40 |
| Radiation [Å] | 1.5418 (CuKα) |
| Divergence slit [mm] | 0.165 |
| Primary soller slit [°] | 2.5 |
| Secondary soller slit [°] | not present |
| Measuring range [°2θ] | 15-70 |
| Step-size [°2θ] | 0.02 |
| Counting time [s] | 0.85 |
| Sample rotation [rpm] | 15 |

**Table 1. X-ray diffraction measurement settings**

### 3.3 Micro-X-ray fluorescence

For additional screening of diagenetic alteration, concentrations of major and trace elements (including Fe, Mn and Sr) in
three specimen of *A. benedeni benedeni* (SG-125, SG-126, SG-127) were determined using non-destructive, energy-



dispersive micro-X-ray fluorescence (µXRF) analysis. A benchtop Bruker M4 Tornado µXRF scanner was used for this, available at the Analytical, Environmental, and Geochemistry Research Group (AMGC) of the Vrije Universiteit Brussel (Brussels, Belgium). The Bruker M4 Tornado is equipped with a 30 W Rh anode metal-ceramic X-ray tube operated at 50 kV and 600 µA. A polycapillary lens focuses an X-ray beam on the polished sample surface on a spot with a diameter of 25

µm (calibrated for Mo kα radiation). Fluorescing X-ray radiation is detected using two 30 mm2 silicon drift detectors with a spectral resolution of 135 eV (calibrated for Mn-kα radiation). The system is operated under near-vacuum conditions (20 mbar) to allow lower energy radiation of lighter elements (e.g., Mg) to be measured. More details on the µXRF measurement setup and acquisition are provided in de Winter and Claeys (2017) and Kaskes et al. (2021).

X-ray spectra were produced using a spatial resolution of 25 um and a dwell time of 60 seconds. This setting allows for

enough time to reach the Time of Stable Reproducibility and provides the optimal balance between longer measurement time (resulting in better defined XRF spectra) and higher sample sizes (see discussion in de Winter et al. 2017b). Spectra were quantified using on-line calibration with the matrix-matched standard BAS-CRM393 (Bureau of Analyzed Samples, Middlesborough, UK) in the M4 software (following de Winter et al. 2017b). After quantification, the trace element results were calibrated using an off-line calibration constructed using a set of matrix-matched carbonate standards (CCH-1;

Université de Liège, Belgium, COQ-1; US Geological Survey, Denver, CO, USA, CRM393, CRM512, CRM513, ECRM782; Bureau of Analyzed Samples, Middlesborough, UK; and SRM-1d; National Institute of Standards and Technology, Gaithersburg, MD, USA; see e.g. de Winter et al. 2021). All major and trace element data can be found in the supplementary materials (see data availability).

### 3.4 Electron backscatter diffraction

To determine which microstructures are present in *A. benedeni benedeni* and to identify the potential occurrences of minor, localised diagenetic calcite that was not detected through XRD or XRF analysis, electronic backscatter diffraction (EBSD) was carried out on thin sections (ca. 40 µm) of all three specimens. EBSD involves the diffraction of electrons, generated by a SEM-based electron beam, off the crystal planes of a sample, forming a diffraction pattern related to the crystal lattice parameters. This allows for the precise determination of crystal orientation, shape, and size (e.g. Cusack 2016). EBSD

analysis was carried out using an Oxford Instruments Symmetry EBSD detector attached to a Zeiss Gemini 450 SEM at Utrecht University. Thin sections of the samples were mechanically polished with 0.3 µm aluminium oxide suspension and finished with chemical Syton® polish. The thin sections were covered with a thin (several nm) carbon coating to keep charge build-up during the measurements at a minimum. Beam and map acquisition settings (voltage, beam current, dwell time, step size, high/low vacuum) were varied to obtain the optimal results for each map; the acquisition settings for the map shown

here are indicated in Fig. 5. Data processing was done in the Oxford Instruments AZtecCrystal software. Data clean-up consisted of wild spike removal followed by filling in unclassified areas using an iterative nearest neighbour procedure from eight down to six nearest neighbours. The map presented here had 90% indexing after this step. The raw data of this map, as well as additional maps that are not presented here, can be found in the supplementary materials (see data availability). The




grain boundary threshold was set to 10°. Contoured pole figures were plotted to visualise the main crystallite orientations
(Fig. 5f). For these pole figures, multiples of uniform density (MUD) values were calculated. The MUD value is an
indication of the degree of clustering of the poles relative to a random distribution: the higher the MUD, the stronger the co-
orientation (or preferred orientation) of the crystals.

### 3.5 Stable isotope analysis

### 3.5.1 Sample preparation

Samples of approximately 100-300 μg were taken from the outer surface of two specimens (SG-126 and SG-127) by drilling
into it from the exterior along commarginal paths. A Dremel model 225 hand-held drill was used, which was equipped with a
tungsten carbide drill bit of 0.8 mm in diameter (Fig. A1). During sampling, care was taken not to drill too deep as to not
drill into the inner shell layers. The distance from the umbo to the sample track at the axis of maximum growth was
measured in planar view with digital callipers (Fig. A1). The entire shell height of specimen SG-126 was sampled (Fig. A1
a). A total of 55 samples was taken from specimen SG-126 and a total of 30 samples was taken from specimen SG-127 (Fig.
A1 b). The sampling resolution was 0.10 to 1.07 mm (mean: 0.48 mm, 1σ: 0.24 mm; no significant difference between the
two specimens, p>0.05).

### 3.5.2 Oxygen isotope mass spectrometry

Drilled samples from specimen SG-126 were first analysed for carbonate oxygen isotopes ($\delta^{18}O_c$) on a Thermo Scientific
GasBench II gas preparation system coupled to a Thermo Scientific MAT 253 isotope ratio mass spectrometer. 50-100 μg of
powdered sample material was weighed on a Sartorius microbalance and placed in glass vials that were sealed off with
barrier septa. 50-100 μg of two types of in-house standards, Naxos marble and Kiel carbonate, was weighed and placed in
vials as well. Their accepted values are given in Table A1. Each sample was then flushed with helium for 5 minutes to
remove atmospheric air. The samples were reacted with 103% phosphoric acid at 70°C. The produced CO2 gas was led
through a series of cleaning vessels, consisting of a first water trap, a gas chromatograph, and a second water trap. It then
entered the mass spectrometer and masses 44, 45, and 46 were measured using a LIDI (long-integration dual-inlet) workflow
(Hu et al. 2014). A Naxos standard was measured approximately every ten samples during the run. From the bivalve
samples, every tenth sample was weighed in duplicate. These duplicates were measured at the end of the run. The carbon
isotope data ($\delta^{13}C_c$) are reported only when it is relevant for the reproducibility of the analyses. All stable isotope GasBench
data can be found in the supplementary materials (see data availability). Sample $\delta^{18}O_c$ data were corrected for intensity-
dependent fractionation using a linear regression between $\delta^{18}O_{NAXOS}$ and mass 44 intensity (Fig. A2). No significant change
in the Naxos standard values over time (also called "drift") was observed during the measurement. The $\delta^{18}O_c$ and $\delta^{13}C_c$ data
are reported relative to VPDB.




### 3.5.3 Clumped isotopes mass spectrometry

Due to the near-equilibrium precipitation of bivalves, we can use clumped isotope analysis to reconstruct seasonal water temperatures from their carbonate shells (Huyghe et al. 2022). Clumped isotope analysis on carbonate involves analysing the abundance of $^{18}$O-$^{13}$C bonds and has several advantages compared to conventional oxygen stable isotope ($\delta^{18}O_c$) temperature reconstruction. The deviation of the occurrence of these heavy $^{18}$O-$^{13}$C bonds from the expected stochastic distribution is thermodynamically determined and thus independent of the isotopic composition of oxygen of the seawater ($\delta^{18}O_{sw}$) and

carbon of the DIC ($\delta^{13}$C) (e.g., Ghosh et al. 2006; Schauble et al. 2006). This deviation, termed $\Delta_{47}$, is given in per mil and defined as follows:

$$\Delta_{47} = \left[ \left( \frac{R^{47}}{R^{47*}} - 1 \right) - \left( \frac{R^{46}}{R^{46*}} - 1 \right) - \left( \frac{R^{45}}{R^{45*}} - 1 \right) \right] \times 1000 \tag{1}$$

in which $R^n$ is the abundance ratio of isotopic mass n over mass 44 (the most abundant mass, formed by $^{12}C^{16}O^{16}O$) in the sample relative to a standard, and $R^{n*}$ is the theoretical abundance ratio of mass n over mass 44 in the sample relative to a standard if that sample were to have a stochastic distribution (Eiler 2007). Since the value of $\Delta_{47}$ depends only on formation temperature, it allows precise temperature reconstruction without knowledge of $\delta^{18}O_{sw}$, which is required for the more traditional palaeothermometer based only on the $\delta^{18}$O of shells ($\delta^{18}O_c$). This independence from $\delta^{18}O_{sw}$ is especially

important for the mid-Piacenzian Warm Period, as the size of the ice sheets—which store light $^{16}$O and thus drive up the $\delta^{18}$O of the oceans—is not well constrained for this period (e.g., Dowsett et al. 2016). It is also relevant for the shallow-marine waters in which bivalves often live, as the $\delta^{18}O_{sw}$ can be influenced by changing influxes in isotopically light river water (e.g., Schöne et al. 2004; Chauvaud et al. 2005; Johnson et al. 2009)

Clumped isotope values ($\Delta_{47}$) were analysed using a Kiel IV carbonate device coupled to a Thermo Scientific MAT 253 Plus

isotope ratio mass spectrometer (available at Utrecht University) with a LIDI workflow (Hu et al. 2014; Müller et al. 2017) and following the protocol described by Meckler et al. (2014). For specimen SG-126, analysis was focussed on samples with $\delta^{18}O_c$ values in the ranges 2.0-3.3‰ (close to the maximum) and 0.4-1.2‰ (close to the minimum), based on the previously measured samples (sect. 3.5.2). From specimen SG-127, all samples were analysed. "Sample" here refers to the total amount of powder drilled from a single sampling track (Fig. A1), whereas "aliquot", "measurement", or "datapoint" refers to a small

amount of powder taken from a sample and measured for stable isotopes. Multiple aliquots were measured from each sample. 75-95 μg of powdered material was weighed with a Mettler Toledo microbalance or a Sartorius microbalance and placed in glass vials. Similarly weighed carbonate standards were measured in an approximately one-to-one ratio to the standards in each run (22 samples, 24 standards; Kocken et al. 2019). These standards were two control standards—Merck CaCO$_3$ (synthetic; product code 1.02059.0050) and IAEA-C2 (Bavarian travertine)—as well as ETH-1, ETH-2, and ETH-3

(Bernasconi et al. 2018; 2021). Their composition is given in Table A1. The ETH standards, which have varying $\Delta_{47}$, $\delta^{18}O_c$, and $\delta^{13}$C compositions, were used to transfer the sample $\Delta_{47}$ values to the I-CDES reference frame (Bernasconi et al. 2021).



Merck and IAEA-C2 were not involved in data processing (treated as samples) and used to assess long-term measuring uncertainty.

Carbonates were reacted with 105% phosphoric acid at 70°C. The produced CO2 gas was purified through two cryogenic
traps (-170°C) and a Porapak trap (-50°C). It then entered the mass spectrometer and masses 44-49 were measured against a reference gas of known composition ($\delta^{18}O = -4.67‰$, $\delta^{13}C = -2.82‰$). A negative baseline correction proportional to the mass 44 intensity was performed for all masses. The calculated δ values were then corrected for $^{17}O$ through the method presented in Brand et al. (2010). The $\Delta_{47}$ values were calculated and averaged over the 40 pulses as per the LIDI system. Clumped results were corrected for drift through bracketing with ETH-3 standards. An empirical transfer function (ETF) was
constructed by regressing the raw ETH-1, ETH-2, and ETH-3 values over their accepted $\Delta_{47}$ values (Bernasconi et al. 2021) and using the resulting linear regression line to transfer the sample $\Delta_{47}$ values to the I-CDES90°C reference frame (Fig. A3). For the ETF, 403 ETH standards from multiple runs measured over the span of several weeks were used for linear regression (53 ETH-1, 64 ETH-2, 286 ETH-3; Fig. A4). No acid fractionation correction was necessary even though the samples were reacted at 70°C, as this offset is already incorporated into the new values of the ETH standards in the I-CDES reference
frame (Bernasconi et al. 2021).

The $\delta^{18}O_c$ and $\delta^{13}C$ data of the same measurements were corrected through a 15-point running average to eliminate long-term trends present within the $\delta^{18}O_c$ and $\delta^{13}C$ raw data. Mass-dependent fractionation occurred in some samples, likely due to the loss of a fraction of the CO2 through leakage. This is evident from a correlation between $\delta^{18}O_c$ and $\delta^{13}C$ (these runs are marked in Fig. A4). The $\delta^{18}O_c$ and $\delta^{13}C$ data from these runs were not used for the running average correction nor were
they used in the $\delta^{18}O_c$ records of *A. benedeni benedeni*, as their isotopic composition no longer reflected the original signal. As $\Delta_{47}$ is calculated as the deviation from a stochastic distribution for a given $\delta^{18}O_c$ and $\delta^{13}C$ composition, it was not influenced by this fractionation. The final $\delta^{18}O_c$ and $\delta^{13}C$ values were reported relative to VPDB. Samples that showed a strong drift during the 40 LIDI measurement pulses, had a low intensity, a high standard deviation, or showed signs of contamination were deemed unreliable and were removed from the dataset. The intensity cut-off was <9.0 V. The cut-off for
standard deviations in $\Delta_{47}$ data within the 40 measurement pulses was >0.10‰ for both standards and samples. Contamination evident from increased intensity on the mass 49 cup, was quantified using the 49 parameter—the ratio between mass 49 and mass 44 intensities—and a cut-off of >0.1 was used for standards, and >0.2 for samples. All standards used for correction, as well as the IAEA-C2 and Merck values and the samples, can be found in the supplementary materials (see data availability). After removing erroneous samples, 103 sample measurements remained.

To obtain warm vs cold datasets from the clumped isotope data, the samples were grouped based on their $\delta^{18}O_c$ values, with low values corresponding to warm temperatures and vice versa. The $\Delta_{47}$ data from the same aliquots were then averaged to obtain warm and cold average temperatures. As individual $\Delta_{47}$ measurements have a large uncertainty, $\Delta_{47}$ datapoints cannot be split into meaningful "warm" and "cold" groups directly. The choice of a cut-off value for warm and cold $\delta^{18}O_c$ values is a trade-off between confidence and seasonality: A cut-off close to the average $\delta^{18}O_c$ value results in a higher sample size and
a narrower confidence interval for the resulting temperature reconstruction. Such a cut-off, however, also results in averaging





samples from a large part of the year and thus a dampening of the seasonality range. For a cut-off that only includes the lowest and highest $\delta^{18}O_c$ values, the opposite is true. To find the best compromise between confidence and seasonality, the average warm and cold $\Delta_{47}$ values and their associated 95% confidence levels were plotted for a range of $\delta^{18}O_c$-based cut-offs (see also Fig. 9). A Student's t-test was used to determine whether cold and warm datasets were statistically different at

a 95% confidence level. Instead of looking at each *A. benedeni benedeni* specimen individually, $\delta^{18}O_c$ data from both specimens were combined in order to improve the statistics.

**3.6 Palaeotemperature reconstruction**

$\Delta_{47}$ values were converted to temperatures using the temperature transfer function of Meinicke et al. (2020; 2021):

$$\Delta_{47}[‰\ ICDES_{90°C}] = 0.0397 \pm 0.0011 * 10^6 * T^{-2}[K] + 0.1518 \pm 0.0128 \tag{2}$$

This temperature transfer function is based on foraminifera and therefore more suited to the relatively low temperature range in which bivalves live. Since this function for $\Delta_{47}$ is not linear, errors associated with the $\Delta_{47}$ data were propagated using a Monte Carlo simulation (N = $10^5$) of uncertainty in slope and intercept of the temperature transfer function as well as

measurement uncertainty on $\Delta_{47}$ values assuming a normal uncertainty distribution. The standard deviation, standard error, and 95% confidence level were then calculated from this normally distributed simulated temperature dataset. All calculations can be found in the supplementary materials (see code availability). Uncertainties on temperatures were calculated after temperature conversion. Since the temperature transfer function for clumped isotopes is not linear, this introduces a bias, specifically towards warmer temperatures. Therefore, the temperature means were calculated from averaging the $\Delta_{47}$ values

instead, and the Monte Carlo-generated errors were transferred to these mean temperatures.

The $\delta^{18}O$ of the seawater ($\delta^{18}O_{sw}$) was calculated from $\Delta_{47}$-based temperatures and $\delta^{18}O_c$ data using the temperature transfer function for aragonite of Grossman and Ku (1986), modified by Dettman et al. (1999):

$$\delta^{18}O_{sw}\ [‰VSMOW] = \delta^{18}O_c\ [‰VPDB] - \frac{20.6 - T[°C]}{4.34} + 0.2 \tag{3}$$


This average $\delta^{18}O_{sw}$, the average $\Delta_{47}$-based temperature, and minimum and maximum $\delta^{18}O_c$ values were then re-inserted into this equation to obtain $\delta^{18}O_c$-based summer and winter temperatures. This was done as there were not enough clumped isotope data to reconstruct $\Delta_{47}$-based summer and winter temperatures (see section 4.4.3 and 4.4.4).

**3.7 Growth models**

To gain insights into the nature of the growth of *A. benedeni benedeni*, two types of growth models were fitted to the growth data of specimen SG-126: The Von Bertalanffy growth function (VBGF), a logarithmic model, and the Gompertz equation, a



logistic model. Both are widely used for estimating and analysing growth in modern species (e.g., Lee et al. 2020). The built-in nls() function in R (R Core Team 2021) was used to fit the data to both growth functions (see code availability). This function calculates the nonlinear least-squares estimates of the parameters of a nonlinear model that is defined by the user.

The VBGF and the Gompertz equations are restrictive in terms of how much their shape can change, and much more sophisticated approaches are available today (Lee et al. 2020). However, as this study has only one specimen available for growth model fitting. and since it concerns a fossil species, we believe that this simple approach is appropriate.

## 4 Results

### 4.1 Light microscopy

#### 4.1.1 Structural analysis

The material of all three specimens looks pristine: the narrow increments are sharp and clearly visible, and there are no signs of dissolution or recrystallization (Fig. 2a-c). Four layers are visible in each specimen (Fig. 2a, d). These layers are named




following Bieler et al. (2014), where M+ and M- denote layers external and internal relative to the pallial myostracum (M).

Figure 2: Digital microscopy images of shell layers in the *A. benedeni benedeni* shells. Layers are marked 1-4, growth increments are indicated with dark blue dotted lines, dog = direction of growth. (a) section of SG-127 indicating all four layers, magnification of x250. 1: layer M+2 with crescent-shaped growth increments. 2: layer M+1 with low-angle growth increments and prism-like crystals parallel to the layer thickness. 3: layer M-1 with horizontal, parallel growth increments. 4: layer M-2, dark brown with horizontal, parallel growth increments (difficult to see in image a due to a similar tone to the background). (b) section of SG-126 showing a part of layer M+2 in more detail, magnification of x500. (c) section of SG-126 showing a part of layer M-2 at a better contrast, magnification of x250. (d) schematic overview of the four shell layers and their positions relative to each other and the pallial myostracum (M). In brackets is the equivalent of the layer name in the terminology of Popov (1986).



Their equivalent to another common naming scheme (e.g., Popov 1986; 2014; Milano et al. 2017) is shown in Fig. 2d. Starting from the outer surface, these four layers are: 1) a thin (ca. 100 μm) white-to-grey coloured layer with crescent-shaped growth increments, termed M+2; 2) a ca. 200 μm thick layer (measured at the thickest point, halfway along the shell height) with low-angle growth increments, a white-beige colour, and a prism-like structure that is oriented parallel to the layer thickness, termed M+1; 3) a thick (ca. 1000 μm) layer with clear growth increments parallel to the layer boundaries and a white-beige colour, termed M-1; and 4) a thin (ca. 50 μm) brown-coloured layer with horizontal, parallel growth increments, termed M-2. Layer M+2 extends from the ventral margin to near the hinge, where it thins and ultimately disappears. Layer M+1 thickens toward the ventral margin. Layer M-1 forms most of the hinge. It thins toward the ventral margin. Layer M-2 covers the inside of the shell. It starts at the umbo and extends to the ventral margin.

### 4.1.2 Growth increments

In specimen SG-126, a maximum of 115 growth lines was counted in layer M-1 at the hinge (Fig. 3a, c). In specimen SG-127, a maximum of 131 growth lines was counted just ventral of the hinge (Fig. 3b, c), also in layer M-1. The growth lines were not all clearly visible, and in some areas in the shell, less growth lines could be discerned. As it was deemed easier to miss some growth lines due to the limited resolution than to overestimate the number, the microscope images with the maximum counted number of growth lines are shown here, as they are thought to be the best estimate of the actual number of growth lines.





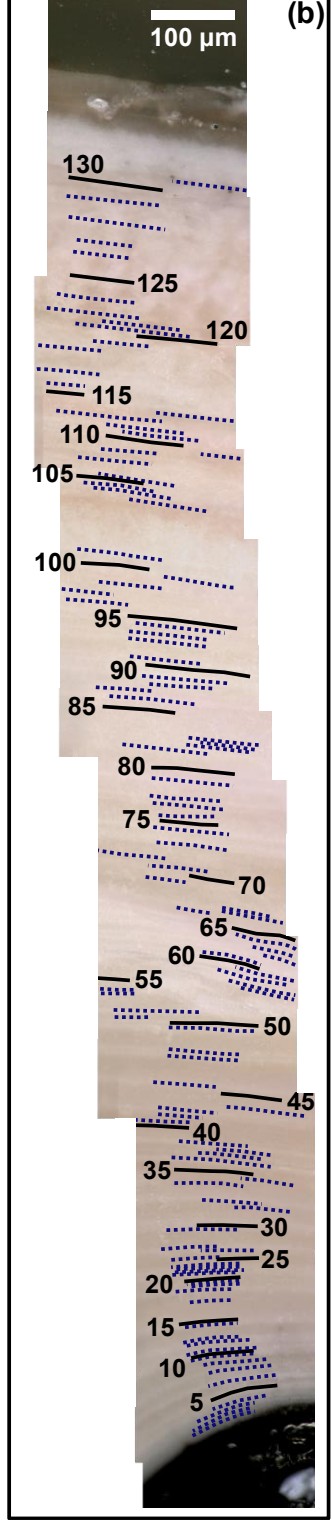




**Figure 3: Growth lines in the *A. benedeni benedeni* specimens. (a) Counted growth lines in specimen SG-126, 115 in total. (b) Counted growth lines in specimen SG-127, 131 in total. (c) Schematic figure of the shell indicating where the growth lines were counted.**

## 4.2 X-ray based analyses

X-ray diffraction analysis (XRD) and micro-X-ray fluorescence (µXRF) point analysis indicate that the shells of *A. benedeni benedeni* were not diagenetically altered. XRD analysis revealed that the shell of *A. benedeni benedeni* consists of 100% original aragonite, as its spectrum is identical to that of pure aragonite (Fig. 4a). Micro-X-ray fluorescence (µXRF) point analyses following the methodology in de Winter et al. (2017b) supports this interpretation, as the specimen is high in Sr, while low in Fe and Mn (Fig. 4b). Higher concentrations of Fe and Mn and lower concentrations of Sr in fossil carbonates are generally associated with diagenetic alteration (e.g., partial recrystallisation) of the carbonate (e.g., Brand and Veizer 1980). Pore waters in many common burial environments (especially in shallow marine successions) become more reducing over time due to the decay of buried organic matter in the absence of oxygen (Calvert and Pedersen 1993). These conditions tend to remobilise Mn and Fe, which are fixed in oxides under surface conditions, resulting in high Fe and Mn and low Sr concentrations compared to those in seawater. These distinct pore water concentrations are captured in recrystallised carbonates during diagenesis, causing high Mn and Fe concentrations and low Sr concentrations to serve as indicators for burial alteration (Al-Aasm and Veizer 1986; Hendry et al. 1995). The absence of high Fe and Mn concentrations and presence of high Sr suggests that this process has not taken place in this specimen.

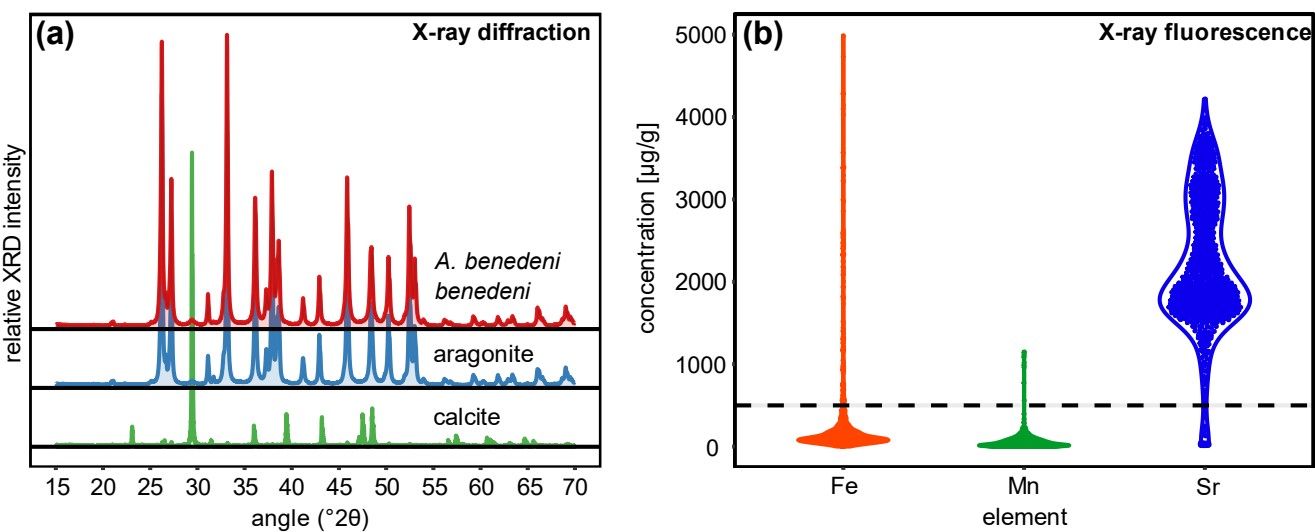

**Figure 4: X-ray based analysis results to assess diagenetic alteration in the specimens. (a) X-ray diffraction analysis of *A. benedeni benedeni* compared with patterns of pure aragonite and calcite, indicating an aragonitic composition of the shell. (b) Violin plot showing the results of the X-ray fluorescence analysis of *A. benedeni benedeni* for the elements Fe, Mn, and Sr. The "violin" shape depicts the kernel density function for the elemental concentration, plotted vertically and mirrored. The specimen is high in Sr and low in Fe and Mn.**




### 4.3 Electron backscatter diffraction

#### 4.3.1 Diagenesis assessment

Electron backscatter diffraction (EBSD) analysis shows no evidence of diagenetic alteration in the shells. In the EBSD maps, 0 to 0.2% of the area was classified as calcite. Upon inspection, these "calcite" grains are actually wild spikes that were subsequently removed in the data clean-up (Fig. B1). There was no local secondary mineral growth visible in any map (e.g., Fig. 5b-e), which would have presented itself as larger crystals ("blocky calcite") that do not follow the surrounding structures (Cusack 2016; Casella et al. 2017). The pole figures show a clear preferred orientation (Fig. 5f), which is

indicative of original growth structures (Casella et al. 2017).

#### 4.3.2 Grain size and orientation

The aragonitic shell material is very fine grained and its crystals show a strong preferred orientation as well as twinning. Most grains have an area of a few $\mu m^2$ or smaller (Fig. 5g). Layer M-1 has a narrower grain size distribution than the outer and inner outer layers, with no grains larger than 25 $\mu m^2$ present in the analysed section. The multiples of uniform density

(MUD) values indicate a strong preferred orientation (Fig. 5f). Layer M-1 has an increased MUD of 104.07 compared to ca. 40 for the other two layers, indicating a stronger preferred orientation (Fig. 5f). This is mainly due to the strong preferred orientation of the 100 axis, while the 001 and 010 axes show a weaker preferred orientation compared to the other layers. The pole plots show double maxima of the 001 and 010 axes, which rotate around the 100 axes (M+2 and M+1, Fig. 5f). This is observed in layers M+2 and M+1. In layer M-1, the 001 and 010 axes form a girdle around the 100 axis rather than

two distinct maxima (M-1, Fig. 5f). The rotational 100 axis is generally oriented parallel to the growth direction. This is especially evident from the lamellae in layer M-1 of Fig. 5b-e.

The angle of ca. 64° between the 001 and 010 axes is characteristic of aragonite polysynthetic (110) twinning (e.g., Griesshaber et al. 2013). Calculated twinning boundaries indicate that the two dominant orientations, as indicated by the IPF colour coding, in layers M+2 and M-1 represent twins (Fig. 5i). In layer M+1, the two dominant orientations that make up

the lamellae are not a result of twinning. Instead, each of the two orientations (orange and purple) consists of their own set of twins (light and dark orange and purple, respectively; Fig. 5i).











**Figure 5: EBSD map of specimen SG-127. Map acquisition specifics: 20 kV, high vacuum, 0.2 µm step size. (a) Band contrast map with scale, orientation, growth lines, direction of growth (dog), and shell layers M+2, M+1, and M-1, separated by striped lines. (b)**
**EBSD IPF X map, showing grain orientations parallel to the X0 axis. Unit cells of aragonite with the most common orientations are shown, the colour of which corresponds to the colour coding of the map. (c) EBSD IPF Y map showing grain orientations parallel to the Y0 axis, with unit cells. (d) EBSD IPF Z map showing grain orientations parallel to the Z0 axis, with unit cells. (e) Inverse pole figure (IPF) colour key for the EBSD orientation maps. (f) Contoured pole figures for the three main crystallographic directions and the MUD, for the different shell layers. (g) Grain size distribution chart for the entire map and the different shell**
**layers. Note the logarithmic y-axis. (h) Schematic figure indicating the approximate location of the map on the shell. (i) Shell layers from map IPF Z zoomed in and with added grain (black) and twinning (yellow) boundaries.**

### 4.3.3 Microstructures

Within the EBSD maps, several different structures are observed (Fig. 6). Layer M+2 shows bundles of crystals that diverge from the centre of the layer and show overlap in a scale-like pattern (Fig. 6a). Layer M+1 shows two sets of crystal
orientations that are oriented perpendicular to the growth lines and that alternate and interfinger, similar to a zebra pattern (Fig. 6b). Layer M-1 shows various structures: In the hinge region, a hatched pattern is visible (Fig. 6c), In other parts of M-1, elongated, columnar crystals with their long side oriented parallel to the growth direction are observed (Fig. 6d). Around this prismatic structure, there are small grains with no clear preferred shape orientation (Fig. 6d).

### 4.4 Stable isotope analysis

### 4.4.1 Oxygen isotopes

The $\delta^{18}O_c$ data of specimens reveal sinusoidal records that range between +0.3 and +3.3‰ (Fig. 7a, b) and suggest multi-annual growth, with $\delta^{18}O_c$ peaks corresponding to winters and valleys corresponding to summers. Specimen SG-126 shows at least 8-9 years of growth (Fig. 7a). The first and last years of this specimen may not have been recovered during sampling, as the first and last few millimetres were not sampled. In addition, the sampling resolution was likely too coarse to capture
all years in the ontogenetically oldest section of the shell, as growth increments generally become thinner with age. In SG-126, high $\delta^{18}O_c$ values generally correspond to darker bands and vice versa, which suggests that the dark-light couplets represent the winter and summer seasons. The $\delta^{18}O_c$ variability decreases near the ventral margin of the shell.

The sampled section of SG-127 shows six cycles in $\delta^{18}O_c$ values (Fig. 7b). Contrary to SG-126, the relationship between $\delta^{18}O_c$ value and shell colour is not as straightforward. The dark-light alternation on SG-127 is not as consistent as the pattern
seen on SG-126. There are many very thin light and dark bands on SG-127, and many of the samples likely contain material from both light and dark bands (Fig. 7b; not all indicated on Fig. 7b as the exact location relative to the samples was not determined).

The internal reproducibility of conventional stable isotope analyses was 0.04‰ (1σ) for $\delta^{13}C$ and 0.06‰ (1σ) for $\delta^{18}O_c$. External reproducibility was 0.05‰ for $\delta^{13}C$ and 0.07‰ for $\delta^{18}O_c$ based on repeated measurements of the Naxos standard.
The Kiel carbonate standard showed external reproducibility of 0.09‰ for $\delta^{13}C$ and 0.06‰ for $\delta^{18}O_c$.





**Figure 6: Different shell structures identified in the EBSD maps. (a) Overlapping scale-like structure in layer M+2. (b) Alternating zebra-like patterns in layer M+1. (c) Hatched pattern in layer M-1 at the hinge of the shell. (d) Rectangular, columnar crystals with their length oriented perpendicular to the growth lines in layer M-1. There are also areas in this layer (e.g., top right in panel d) without a clear grain shape structure. (e) Schematic figure of the shell indicating the approximate location of the maps.**




_(a) SG-126 δ¹⁸Oc and (b) SG-127 δ¹⁸Oc profiles of δ¹⁸O [‰ VPDB] versus distance from umbo [mm], with Average, 1σ, 2σ, Dark growth bands on shell, and Light growth bands on shell indicated.)_





**Figure 7: $\delta^{18}O_c$ records of the *A. benedeni benedeni* shells, plotted relative to the distance from the umbo. (a) Record of specimen SG-126, consisting of 55 samples each measured 1 to 3 times. (b) Record of specimen SG-127, consisting of 30 samples. The average was calculated from the different measurements of a single sample. If there was just one measurement, that value was**
**taken. One and two standard deviations are plotted around the average. The standard deviation was calculated as described in sect. 4.4.1 (0.15‰). In the background, the approximate colour of the growth band on the outer surface of the shell is indicated (see also Fig. A1). The warm and cold $\delta^{18}O_c$ 'bins' are marked in red and blue bands. These bins encompass all datapoints below 0.9‰ and above 2.4‰. See also section 3.4.3 and Fig. 9.**

The external reproducibility of the standards, however, does not reflect the reproducibility of the samples based on duplicate
measurements. The samples from each sampling track were not homogeneous, as they likely contain multiple thin growth increments and were not homogenised after drilling. Therefore, different aliquots from the same sample sometimes resulted in quite large differences when measured, in both the GasBench and the MAT 253 PLUS analyses. Instead of using the external reproducibility on the Naxos standard, the uncertainty on the samples was determined as follows: for each sample of which multiple aliquots were analysed, the standard deviation of these aliquot results was taken, and all these standard
deviations were averaged. To this goal, $\delta^{18}O_c$ data from GasBench and MAT 253 PLUS analyses were combined. This resulted in an uncertainty of 0.15‰ for SG-126. There were not enough duplicate $\delta^{18}O_c$ measurements for SG-127, so the value of 0.15‰ was also applied to this specimen.

### 4.4.2 Growth curves

Growth model fitting suggests a maximum size of *A. benedeni benedeni* of 34-52 mm. A growth curve has been
reconstructed for the specimen using the $\delta^{18}O_c$ record of specimen SG-126 (Fig. 8a-b). This was not done for specimen SG-127 as only a small section of the entire shell was sampled. Figure 8 also shows the fitted Von Bertalanffy and Gompertz growth models. The VBGF has the form of Eq. 4, where $H$ is the body size of the specimen at time $t$ (here: shell height measured as distance from umbo), $H_{asymp}$ is the asymptotic height (i.e., the average theoretical maximum shell height), $k$ is the growth coefficient, $t$ is the time (here: in years), and $t0$ is the theoretical time where $H$ equals 0.


$$H(t) = H_{asymp}(1 - e^{-k(t-t_0)}) \tag{4}$$

Each parameter, its associated standard error (SE), and p-value were calculated. $H_{asymp}$, $k$, and $t0$ were estimated to be
52.35±7.64 SE (p<0.05), 0.09±0.02 SE (p<0.05), and 0.03±0.16 SE (p>>0.05) respectively. The interpretation of growth
coefficient $k$ is problematic, and besides having a very large standard error and low confidence, $t0$ is merely a mathematical artefact that does not have any biological meaning (Lee et al. 2020). $H_{asymp}$ is the estimated average maximum shell height in millimetres that *A. benedeni benedeni* would have reached. The Gompertz equation has the form of Eq. 5:

$$H(t) = H_{asymp}e^{-be^{-ct}} \tag{5}$$






where $H$ is again the shell height at time $t$, $H_{asymp}$ is the average theoretical maximum shell height, and $b$ and $c$ are coefficients. $H_{asymp}$, $b$, and $c$ were estimated to be 34.11±1.98 SE ($p<0.05$), 2.58±0.14 SE ($p<0.05$), and 0.29±0.03 SE ($p<0.05$), respectively. The Gompertz model thus suggests a shorter average maximum shell height of 34.11 mm. The standard error on the residuals is 0.51 for the VBGF and 0.63 for the Gompertz equation. The R2 value for both fits is 0.99, but as both functions are non-linear, this is not a reliable indicator of goodness-of-fit (e.g., Spiess and Neumeyer 2010).

**Figure 8: Growth curve for specimen SG-126. (a) Growth curve constructed by inserting the counted years versus distance-from-umbo datapoints into two different fitting algorithms: the asymptotic Von Bertalanffy growth function and the logistic Gompertz function. (b) δ¹⁸Oc record with the inferred growth years marked by lines connecting to plot (a).**



### 4.4.3 $\Delta_{47}$ and palaeotemperature

A mean annual temperature (MAT) of 13.5±3.8°C has been calculated through averaging all $\Delta_{47}$ values and converting this average to a single temperature, with the 95%CL determined through Monte Carlo error propagation. $\Delta_{47}$-based summer and winter temperatures could not be determined due to a limited amount of data. The summer and winter data were compiled by selecting samples with respectively low and high $\delta^{18}O_c$ values from both specimens, and subsequently averaging the associated $\Delta_{47}$ values (Fig. 9, see also section 3.4.3). However, Fig. 9 shows that to obtain statistically sound summer and winter temperatures from *A. benedeni benedeni*, more clumped isotope measurements are required than analysed here. The $\delta^{18}O_c$ cut-offs for <0.9‰ and >2.4‰ are highlighted to illustrate this. At these cut-off points, the number of datapoints (N=20 and N=21, respectively) is large enough to bring down the 95% confidence level (CL) range somewhat, but the two datasets are not statistically different (p>0.05, Student's t-test), both due to the wide range included in these cut-offs and the large 95%CL range. Increasing the number of clumped isotope measurements decreases the large errors and allows for making the cut-offs narrower so that they better represent the seasonal extremes. That this approach works, is exemplified by the MAT of 13.5±3.8°C. Due to the large number of measurements (N=103), the error on this temperature has been reduced to a reasonable level. The same is possible for seasonal temperatures as long as the dataset is large enough.

The external reproducibility for the clumped isotope analyses was 38 ppm for IAEA-C2 (n=23) and 60 ppm for Merck (n=15). The external reproducibility of IAEA-C2 was used as the standard deviation on $\Delta_{47}$ for the samples, as it is closer in composition to the samples than Merck (Fig. A4). After corrections, most of the $\Delta_{47}$ data range between 0.55 and 0.70‰ for both specimens (Fig. A5).

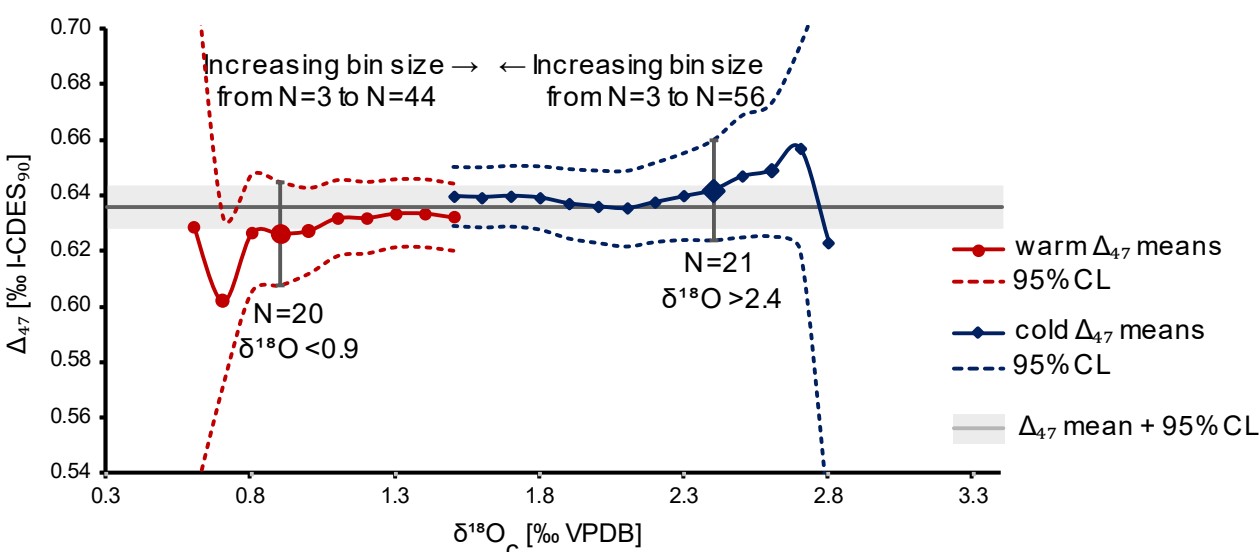

**Figure 9: Overview of averages (round and diamond symbols) and 95% confidence intervals (dashed lines) for different sizes of grouping summer and winter $\Delta_{47}$ aliquots. Sizes of groups increase towards the middle of the plot. The larger symbols with vertical error bars highlight the issue of having insufficient aliquot measurements. The bin sizes are small, and so they should come close to the true (non-averaged) seasonality. However, the number of aliquots is reduced to ca. 20 at this point. This translates to large errors, making these cold and warm averages statistically indistinguishable from each other.**



### 4.4.4 $\delta^{18}O_{sw}$, $\delta^{18}O_c$, and temperature

The average $\delta^{18}O_{sw}$ is 0.10±0.88‰ VSMOW (95%CL), based on the average $\Delta_{47}$-based temperature of 13.5±3.8°C and the average $\delta^{18}O_c$ of both shells combined (1.53±0.015 VSMOW, 95%CL). The Grossman and Ku (1986) equation was applied to the highest and lowest $\delta^{18}O_c$ values and the mean $\delta^{18}O_{sw}$ to obtain new winter and summer temperatures (Fig. 10a). To obtain these highest and lowest $\delta^{18}O_c$ values, the 3 lowest (0.30, 0.42, 0.43‰) and highest (3.07, 3.16, 3.29‰) datapoints were averaged. This resulted in summer and winter temperatures of 18.5±3.9°C and 6.4±3.9°C (95%CL, Fig. 10b).

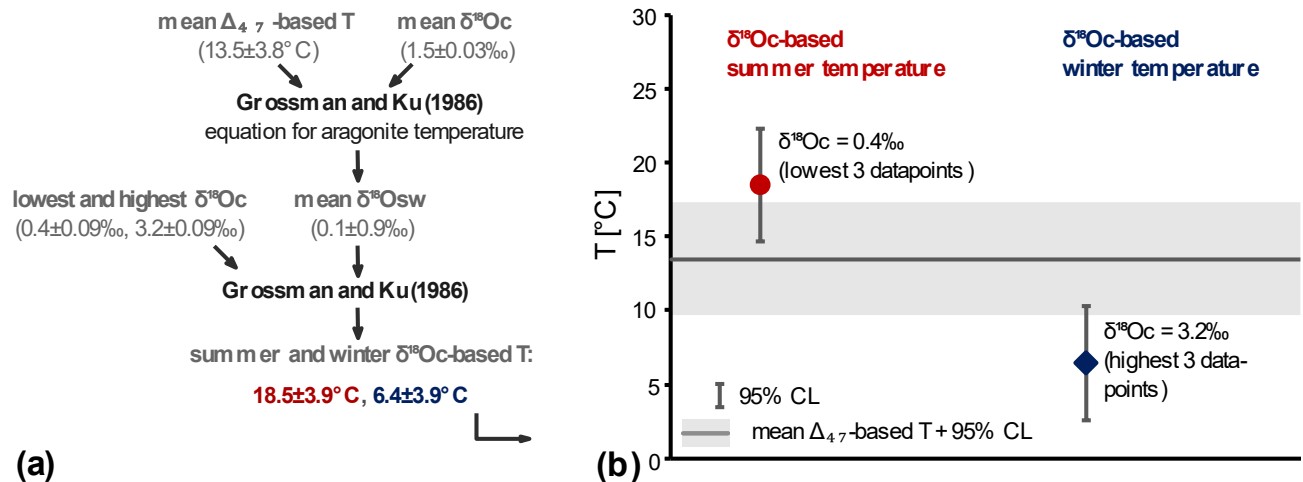


**Figure 10: $\delta^{18}O_c$-based summer and winter temperatures. (a) Schematic illustrating how the $\delta^{18}O_c$-based temperatures have been calculated and what values were used. All values after ± are 95%CLs. b. Mean $\delta^{18}O_c$-based summer (red circle) and winter (blue diamond) temperatures and their 95%CL. In the background, the $\Delta_{47}$-based mean temperature and confidence interval is shown (grey band). The $\delta^{18}O_c$ values used are the average of the lowest and highest 3 datapoints.**

## 5 Discussion


### 5.1 Diagenesis

No signs of diagenesis of the shells were found through micro-X-ray fluorescence (µXRF), X-ray diffraction (XRD), light microscopy, and electron backscatter diffraction (EBSD) analyses. The µXRF analyses indicated that the shell material was high in Sr and low in Fe and Mn, these latter two elements are regularly used to pinpoint diagenesis (Fig. 2b; de Winter &
Claeys, 2016). XRD and EBSD analysis indicated no presence of calcite, to which the metastable aragonite alters as it undergoes diagenesis (e.g., Al-Aasm and Veizer 1986; Casella et al. 2017). The XRD pattern of specimen SG-125 consisted of 100% aragonite, although it must be noted that this specimen was not analysed for stable isotopes. No blocky calcite was observed in the EBSD images of any of the three specimens (Cusack 2016; Casella et al. 2017). This alone does not preclude diagenetic alteration, as calcite is not formed in the earliest stages of diagenesis (Cochran et al. 2010; Marcano et al. 2015;
Ritter et al. 2017) and there is not necessarily a difference in grain size (due to the presence of larger, secondary grains) between pristine and altered aragonite (Casella et al. 2017). Further analysis of the EBSD maps did not, however, produce



any evidence for diagenetic alteration. Microstructures that are similar to those found in modern Tellinidae were observed in the EBSD maps (Popov 2014; see also section 5.2), suggesting that these are original structures. In altered aragonitic bivalves, the structures as observed from EBSD maps are more homogenous and have a lower MUD compared to pristine

material (Casella et al. 2017; study carried out on *Arctica islandica*). The absolute MUD values cannot be compared with different studies due to different measurement settings. However, comparison with maps of pristine versus altered aragonite in Casella et al. (2017) shows that the altered material in Casella et al. (2017) is much more homogeneous and randomised (i.e., low preferred orientation of the grains) than observed in this study. Even the homogeneous structure in *A. benedeni benedeni* (Fig. 6d) appears less chaotic than altered aragonite (Casella et al. 2017). Combined with the shallow burial depth

and the unconsolidated and fine-grained nature of the strata, this evidence suggests that no significant diagenetic alteration took place. The isotope values presented here therefore record the formation temperature of the biogenic aragonite, assuming equilibrium fractionation.

Previous studies have indicated good preservation for bivalves from the Oorderen Member (Valentine et al. 2011; Johnson et al. 2022), from other members of the Lillo Formation (Johnson et al. 2022) and from the early Pliocene Ramsholt Member of

the Coralline Crag Formation (Johnson et al. 2009; Vignols et al. 2019). Pliocene shells from the North Sea basin are expected to yield decent temperature results by geochemical analysis, as long as preservation is not obviously suspected (e.g., in the Sudbourne Member of the Coralline Crag Formation in southeast England, whose aragonitic shells have nearly all been dissolved; Balson et al. 1993).

### 5.2 *Angulus benedeni benedeni* shell structure

The shell of *A. benedeni benedeni* consists of four shell layers, of which the outer three show different macrostructures that are also observed in other tellinid bivalves. Three of the four shell layers can be assigned to a specific bivalve shell structure through comparison with the structures described in the review paper of Popov (2014). The fourth layer, M-2, was not captured on any EBSD map. The bundled, diverging crystals in layer M+2 (Fig. 6a) have been interpreted as a compound composite prismatic structure (Popov 2014). The zebra-like pattern observed in layer M+1 (Fig. 6b) has been interpreted as a

crossed lamellar structure (Popov 2014; Crippa et al. 2020). Layer M-1 shows various structures: the hatched pattern observed in the hinge region (Fig. 6c) is interpreted as a complex crossed lamellar structure (Popov 2014; Crippa et al. 2020). The column-like crystals (Fig. 6d) have been interpreted as a prismatic structure (Popov 2014). The small crystals with no clear preferred orientation that surround this prismatic structure (Fig. 6d) were interpreted as a homogeneous structure (Popov 2014). However, it might also be a different structure that cannot be analysed at this resolution due to the

very fine-grained nature of the aragonite.

Aragonite polysynthetic (110) twinning—indicated by a 64° misorientation angle between the 001 and 010 axes—is frequently observed in bivalves (Kobayashi and Akai 1994; Griesshaber et al. 2013; Crippa et al. 2020) and other molluscs (Schoeppler et al. 2019). It can contribute to rapid shell growth as it is more efficient at filling up space than non-twinned growth (Schoeppler et al. 2019). This twinning is observed in all analysed layers of *A. benedeni benedeni*. As the double 001



and 010 maxima are much stronger in the external layers M+2 and M+1 than in the internal layer M-1, twinning may be more significant in these two layers. This may, in turn, be linked to more rapid growth.

The described microstructures—compound composite prismatic, crossed lamellar, complex crossed lamellar, prismatic, and homogeneous—were all previously described in other tellinid genera such as *Macoma*, *Peronidia*, and the lucinid *Megaxinus* (Popov 2014). However, there is a large variation in structural composition within the Tellinidae family (Popov 2014).

Tellinidae generally have three layers, sometimes with sub-layers present. These three layers as described by Popov (2014) correspond to what is called here M+2 (outer in Popov 2014), M+1 (middle in Popov 2014), and M-1 (inner in Popov 2014); the brown M-2 layer we observed in *A. benedeni benedeni* is not mentioned separately, and is perhaps part of M-1. The M-1 layer in Tellinidae is usually homogeneous, as it looks to be here in some sections. *Angulus benedeni benedeni* appears to be very similar in structure to the closely related *A. nysti*. The latter has an M-1 layer that is complex crossed lamellar with

interlayers of prisms and an M+1 layer that is crossed lamellar (Popov 2014). Various differences are observed as well: layer M+2 of *A. nysti* is fibrous prismatic with megaprisms rather than compound composite prismatic. Furthermore, in *A. nysti* the prisms diverge from the top rather than from the centre of this layer, as in *A. benedeni benedeni* (Popov 2014). Layer M-2 was not studied with EBSD here, but light microscopy suggests that it might be structurally similar to the neighbouring M-1 layer.

This study supports previous investigations in highlighting the utility of EBSD for analysing shell structures in bivalves (e.g., Cusack 2016; Checa et al. 2019; Crippa et al. 2020). As this study documents one of the first EBSD analyses done on fossil molluscs, some suggestions for future research exploring this method include 1) comparing the microstructures of multiple shell layers between species through geological time; 2) producing maps at higher resolution—step size of 0.1 or even 0.05 µm—to better capture very fine-grained microstructures; 3) analysing larger numbers of *A. benedeni benedeni* specimens to

characterise the microstructural variation present within species; and 4) analysing *A. benedeni benedeni* specimens that have undergone diagenesis to determine how that alters the structures.

**5.3 *Angulus benedeni benedeni* growth history**

*Angulus benedeni benedeni* experienced slower growth in winter, could live for up to a decade or longer (Fig. 7, 8), could reach lengths of 34-52 mm, and likely formed monthly growth increments. The maximum (winter) $\delta^{18}O_c$ values decrease,

while the minimum (summer) values stay similar throughout the oxygen isotope record of specimen SG-126. This amplitude reduction may reflect growth breaks during the colder months. This is supported by the correlation of high $\delta^{18}O_c$ values with dark growth bands on the shell's exterior. Darker bands in bivalves often represent high organic matter content related to decreased mineralisation rates (Lutz and Rhoads 1980, cited in Carré et al. 2005). The ontogenetic growth rate decline that is often seen in bivalves (e.g., McConnaughey and Gillikin 2008) is not apparent from the growth curve of specimen SG-126.

The growth curve only becomes slightly less steep in the last few measured years, and there is no marked increase in the wavelength of the $\delta^{18}O_c$ record (Fig. 8).



Growth modelling has provided a preliminary range of maximum shell height for *A. benedeni benedeni*. Both the VBGF and the Gompertz equation show a good fit with the growth data of Shell SG-126, indicated by the similar standard errors of the residuals (0.51 and 0.63 mm, respectively). The data do not appear to show the inflection point that is present in the logistic

Gompertz function but not in the logarithmic VBGF. The exponential growth on the left-hand side of such an inflection is suitable for the first stages of growth in bivalves, when they are still in the larval stage (Urban 2002). Since SG-126 is an adult shell, and since it is likely that not the entire life history was sampled (the first few millimetres of growth may have been missed, see Fig. A1), the larval growth stage is not represented. The second major difference between the VBGF and the Gompertz equation is their estimation of $H_{asymp}$, the average theoretical maximum shell height. The VBGF yields a

maximum shell height estimate of around 52 mm, while the Gompertz equation gives an estimate of around 34 mm. Due to the nature of the respective models, the VBGF tends to overestimate the maximum shell height, while the Gompertz equation tends to underestimate it (Urban 2002), so these two estimates yield a plausible range for the maximum shell height of *A. benedeni benedeni*. The heights of specimens SG-126 and SG-127 (30-36 mm) are close to the $H_{asymp}$ of the Gompertz model, and the other specimens in the collection of the Royal Belgian Insitute of Natural Sciences generally do not exceed

40 mm in length. It should be noted that $H_{asymp}$ represents the average theoretical maximum shell height, and individual shells can grow larger. The shape of the VBGF and Gompertz curves suggest that specimen SG-126 had not yet reached its maximum shell height. The fact that growth had not yet significantly slowed at around 8-9 years suggests that *A. benedeni benedeni* may have lived significantly longer, recording environmental variability on a scale of seasons to decades.

Bivalves can form growth increments on a variety of timescales and their growth rhythm can be influenced by solar and

lunar cycles (e.g., Tran et al. 2011). The number of growth lines counted in *A. benedeni benedeni* were divided by the inferred age of the specimen to determine its growth rhythm. The sampled intervals span several years—around nine years for SG-126, which represents most of the shell height, and around six years for SG-127, which represents only around 1/3rd of the shell height. In both shells, 100+ growth lines have been counted. If it is briefly assumed that most of SG-126's growth years have been captured and using the maximum amount of growth lines counted here (115), approximately 13

growth lines per year are recorded. This corresponds to an interval of approximately 29 days, which matches the periodicity of the monthly synodic lunar-tidal cycle. A crude estimate of the minimum and maximum period is possible based on conservative estimates for the upper and lower limits for the number of counted growth lines (~75 to 150; see section 3.2.3; Fig. 3) and years recorded (~7 to 13, depending on how many early growth years are missing and how much aliasing is present in the ontogenetically oldest part of the shell). These estimates yield a range of a periodicity of 17 days (7 years, 150

growth lines) to 63 days (75 growth lines, 13 years). Both growth lines and years are more likely to be over- rather than underestimated, as their recordings are both limited by either the microscopy or the sampling resolution. Therefore, a 17-day periodicity, which requires a maximum age of 7, is unlikely, making the biweekly tidal cycle an unlikely external forcing candidate. Within this range, the monthly tidal cycle seems the most plausible external forcing for the growth bands in *A. benedeni benedeni*. It is also possible that the formation of growth lines is driven by aperiodic or quasi-periodic processes

such as storms—which would skew the correlation between growth years and growth lines—or is a result of an internally



controlled rhythm rather than being externally forced. Internal forcing is a strong possibility, as there is little difference between the two spring-neap cycles within a monthly cycle (Kvale 2006). If tides exerted influence, the fortnightly spring-neap cycle would be expected to be present as well due to its larger amplitude. Monthly lunar cycles have been observed in bivalves (Pannella and MacClintock 1968), but these are present as bundled spring-neap tide pairs with one being more dominant than the other, not as individual growth lines as observed here. Finally, many bivalves form growth increments on much smaller timescales as well (e.g., (semi-)diurnal or circatidal; Judd, Wilkinson, and Ivany 2018 and references therein). We cannot confidently rule out the presence of such ultradian cyclicity in *A. benedeni benedeni*, and more complete records from multiple specimens may be needed to further characterise and statistically solidify the periodicity in growth lines of *A. benedeni benedeni*.

### 5.4 *Angulus benedeni benedeni* as a climate archive

*Angulus benedeni benedeni* shows promise as an archive for high-resolution climate reconstruction. It can live for up to a decade or longer and enables the reconstruction of multiannual climate records at a seasonal resolution. Both $\delta^{18}O_c$ and $\Delta_{47}$ analyses can be successfully applied to this species, and with a larger dataset, it is possible to obtain clumped isotope-based summer and winter temperatures. This species is especially suited for reconstructions of climate in the Pliocene of the North Sea, where it is common (De Meuter and Laga 1976). However, it dates back to the Late Eocene (Marquet et al. 2008) and can thus be used to reconstruct older climates as well, given that these older specimens are well-preserved. Its relatively thin shell in combination with its long lifespan makes it more challenging to sample at high temporal resolution. This first palaeoclimatological study on *A. benedeni benedeni* highlights the opportunities for using this species as a climate archive. These are not limited to stable isotope analysis, but may extend to, for example, minor and trace elemental analyses by e.g. extended μXRF analyses and Laser-Ablation ICP-MS profiles, that can provide information about short-term changes in the North Sea living environment during the Pliocene (e.g. de Winter et al. 2017a).

### 5.5 Clumped and oxygen isotope based palaeotemperatures

The reconstructed mean annual temperature (MAT) of 13.5±3.8°C based on clumped isotope thermometry is 3.5°C higher than that of the pre-industrial North Sea (Mackenzie and Schiedek 2007; Emeis et al. 2015). Our temperature reconstructions for the mid-Piacenzian Warm Period (mPWP) in the North Sea are similar to a previous estimate of ca. 13°C for the North Sea from the somewhat older shallow marine Coralline Crag Formation in southeast England (Dowsett et al. 2012). In addition, modelled anomalies for the mPWP North Sea area of around +3.5°C are in close agreement (Haywood et al. 2020). Compared to estimates for the global mPWP, a 3.5°C warming is on the higher end of the range given by proxy data (+2-4°C, sea surface temperature, Dowsett et al. 2012) and models (+1.7-5.2°C, surface air temperature, Haywood et al. 2020). It is higher than the modelled average SST warming of +2.8°C (Haywood et al. 2020). The North Sea in the Pliocene appears to have a somewhat heightened sensitivity to increased $CO_2$ concentrations compared to the global average. The North Atlantic showed a much stronger warming of +4-7°C during the mPWP, well outside the global average proxy and model





ranges and indicative of a greater sensitivity (Dowsett et al. 2012; Haywood et al. 2020). The recent increase of 1-1.3°C in North Sea temperatures since the late 1800s (Schöne et al. 2004; Mackenzie and Schiedek 2007; Belkin 2009; Emeis et al.

2015; Quante and Colijn 2016) is similar to the observed increase in global temperatures of 0.9-1.3°C (Gillett et al. 2021). However, most of the warming in the North Sea took place since the 1980s (e.g., Emeis et al. 2015), while the steep warming trend on a global scale took off a few decades earlier, around 1960 (Gillett et al. 2021). Very recent warming in the North Sea thus appears to have been faster than the global average (e.g., Belkin 2009; Quante and Colijn 2016). The current climate is transient, not stable, so a direct comparison is not possible. However, both Pliocene and modern data suggest that

we might see near-future warming in the North Sea that exceeds the global average trend.

This warming was likely not evenly distributed throughout the year during the Pliocene. The $\delta^{18}O_c$-based winter and summer temperatures show a seasonal range of 12.1°C, from 6.4±3.9°C in winter to 18.5±3.9°C in summer (Fig. 10b). This is higher than the modern range of ca. 7-9°C, from 7-8°C in winter to 15-16°C in summer (Kooij et al. 2016). The actual Pliocene range might have been larger, as the calculated range is likely an underestimation due to reasons pertaining to the

methodology as well as to the bivalve-archive and its living environment. Firstly, as the sampling resolution was limited, some of the highest and lowest $\delta^{18}O_c$ values may not have been analysed. Secondly, short growth breaks during winter, as suggested by the dark bands present on the exterior shell surface, may have prevented the coldest temperatures from being recorded in *A. benedeni benedeni*. Thirdly, summer stratification during deposition of the lower Oorderen Member at 40-50 m water depth was found to be a likely possibility by previous research on bivalves from this member (Valentine et al. 2011;

Johnson et al. 2022). Previous research on the somewhat older Coralline Crag Formation also found evidence for summer stratification in the early Pliocene (Jenkins and Houghton 1987; Johnson et al. 2009). When vertical mixing is limited, the summer heat is trapped in the surface layer of the water while the bottom waters remain a cooler temperature. This would lead to underestimation of SSTs in benthic organisms such as bivalves. The upper Oorderen interval from which this study's specimens were collected is interpreted to have been shallower and more strongly influenced by tidal currents than the lower

Oorderen interval from which the shells of Valentine et al. (2011) and Johnson et al. (2022) were collected (Louwye et al. 2004). Still, as the water depth is not well constrained and the thermocline depth is unknown, summer stratification cannot be ruled out for our interval. Finally, the above temperatures assume a constant $\delta^{18}O_{sw}$. A $\delta^{18}O_{sw}$ that varies throughout the year can either dampen or amplify the $\delta^{18}O_c$ signal. A dampening corresponds to a positive correlation between $\delta^{18}O_{sw}$ and temperature, i.e., higher $\delta^{18}O_{sw}$ values in summer, when temperatures are high and $\delta^{18}O_c$ is low, and vice versa. Such a signal

can be caused by enhanced evaporation in the summer, and enhanced precipitation and runoff in the winter/spring. This pattern is expected in a mid-latitude settings such as the North Sea, and a strong positive correlation between $\delta^{18}O_{sw}$ and temperature is observed in the eastern North Sea today (Ullmann et al. 2010). These four factors—limited sampling resolution, possible growth slow-down in winter, possible stratification in summer, and possible dampening due to $\delta^{18}O_{sw}$ fluctuations—all contribute to a potential underestimation of the actual seasonality.

The mPWP seasonal temperature range in the North Sea was likely larger than today, and potentially significantly larger. How much larger remains to be determined from additional clumped isotope data, which can solve the $\delta^{18}O_{sw}$-dampening





issue. The increase in range is attributed to higher summer temperatures and somewhat colder winter temperatures compared to today. This summer-only warming has been observed by previous studies of the Pliocene SNSB (Raffi et al. 1985; Johnson et al. 2009; Valentine et al. 2011; Johnson et al. 2022). In the context of a warmer global climate, this suggests a

reduced heat transport in winter in order to attenuate the overall warming. Such a reduction in heat transport could be related to a change in Gulf Stream intensity, translated to the North Sea via the North Atlantic Current (Valentine et al. 2011).

Rising temperatures in the modern North Sea have already taken their toll on regional ecosystems. A lower primary production has been observed in the past decades, which is partly attributed to warming (Capuzzo et al. 2018). A disturbance of the base of the food web can affect all higher trophic levels and ultimately fisheries. Fish are already affected as well: a

decrease in body size due to rising temperatures was demonstrated by Baudron et al. (2014). While the overall temperature in the North Sea basin has increased, there is no robust evidence that temperatures are rising faster in summer than in winter or vice versa (Quante and Colijn 2016). In the neighbouring Baltic Sea, winter temperatures have been rising faster than summer temperatures over the past few decades (Rutgersson et al. 2014), and asymmetric winter-warming has been predicted for the North Sea area by climate models under RCP scenarios 4.5-8.5 (Quante and Colijn 2016 and references

therein). As the mPWP should not be a one-to-one comparison for modern transient changes, this is not a direct discrepancy. Rather, we should use mPWP data to strengthen climate model predictions. A large part of paleotemperature data in general, and thus of those used for model boundary conditions, reflect an annual average (although some are biased towards one growing season, e.g., $TEX_{86}$ from the Pliocene North Sea; Dearing Crampton-Flood et al. 2020). As such, an important part of the climate system is often missing. Adding explicit summer and winter data has the potential to significantly enrich these

validation datasets. In turn, this will lead to more robust predictions for the near future.

**6 Conclusions**

1.      *Angulus benedeni benedeni*'s macro-and micro shell-structures have been described in detail. Four different macrostructures have been observed. In three of these, three different microstructures were recognized. From the outer to the inner layer, these structures have been identified as complex prismatic, crossed lamellar, and complex crossed lamellar with

prismatic interlayers and possibly homogeneous sections. This is comparable to what has been observed in other tellinid bivalves.

2.      Based on petrography and $\delta^{18}O_c$ measurements, we can state that *A. benedeni benedeni* life's span was up to a decade or more, and it likely formed monthly growth increments. It might have experienced slower growth during winter, as characterised by darker growth bands.

3.      *Angulus benedeni benedeni* shows promise as a climate archive, as demonstrated by the first successful isotope analyses performed on this species. Microsampled oxygen isotope records reveal a characteristic sinusoidal pattern from which summer and winter intervals could be discerned. Clumped isotope analysis revealed that the mean annual temperature in the mid-Piacenzian southern North Sea basin was 13.5±3.8°C, 3.5°C warmer than the pre-industrial average. Preliminary





summer and winter temperatures suggest a larger seasonal range than today, caused by warmer summers but similarly cool
winters. This is in line with several other bivalve-based studies that have found cool winters in the Pliocene North Sea basin.

## Appendices

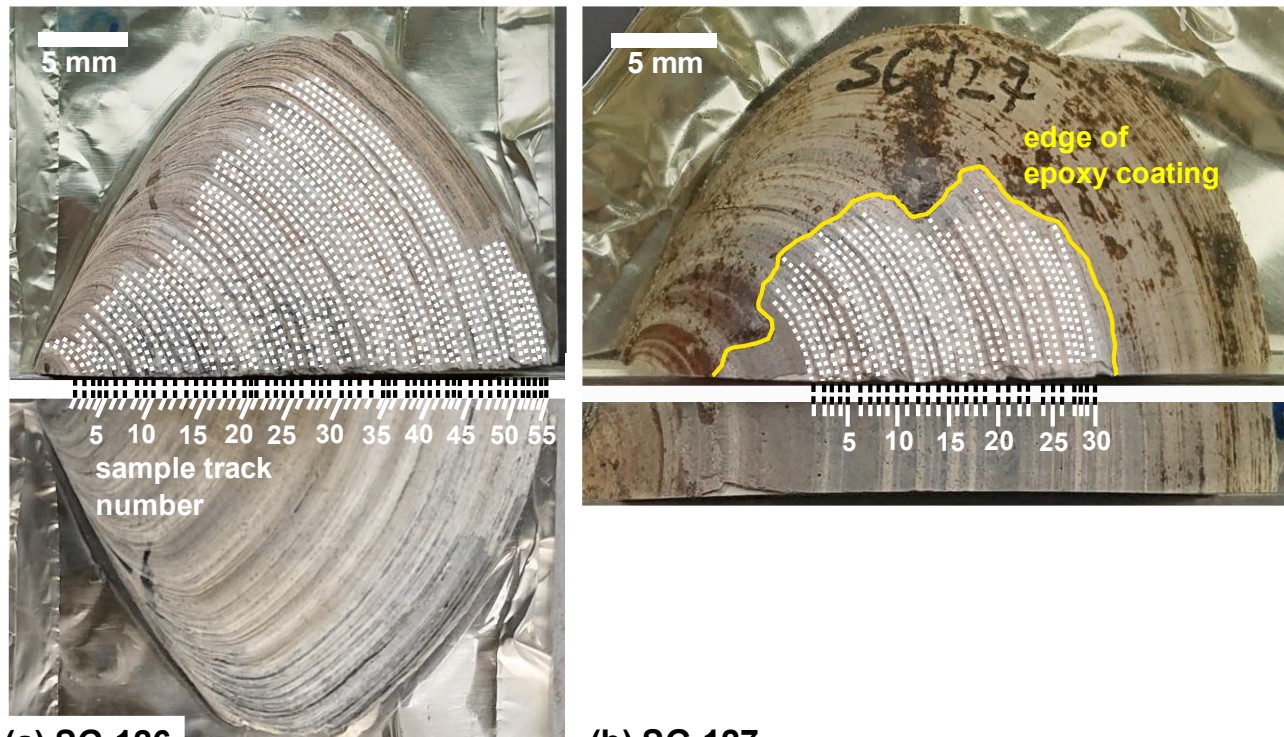

Figure A1: Sampling procedure of the shells for stable isotope analysis. (a) Sample tracks on specimen SG-126, indicated by dotted
lines. 55 in total, each fifth sample track has been labelled. The figure shows the sampled half (top) and that same half prior to
sampling (bottom). (b) Sample tracks on specimen SG-127, indicated by dotted lines. 30 in total. This specimen was for a large part
covered by epoxy, so only a small section at the upper surface could be sampled.

| Standard | Use | $\delta^{18}O$ [‰ VPDB] | $\delta^{13}C$ [‰ VPDB] | $\Delta_{47}$ [‰ I-CDES$_{90}$] |
|---|---|---|---|---|
| Naxos marble | Oxygen isotopes (ThermoScientific MAT 253) | -6.83 | 2.08 | - |
| Kiel carbonate | Oxygen isotopes (ThermoScientific MAT 253) | -16.14 | -35.64 | - |
| ETH-3 | Clumped isotopes (ThermoScientific MAT 253 Plus) | -1.78 | 1.71 | 0.6132 |



| | | | | |
|---|---|---|---|---|
| ETH-2 | Clumped isotopes (ThermoScientific MAT 253 Plus) | -18.69 | -10.17 | 0.2085 |
| ETH-1 | Clumped isotopes (ThermoScientific MAT 253 Plus) | -2.19 | 2.02 | 0.2052 |
| IAEA-C2 | Clumped isotopes (ThermoScientific MAT 253 Plus) | -9.00 | -8.25 | 0.6409 |
| Merck | Clumped isotopes (ThermoScientific MAT 253 Plus) | -15.51 | -42.21 | 0.5135 |

**Table A1: Accepted values for stable isotope measurements, both oxygen and clumped.**

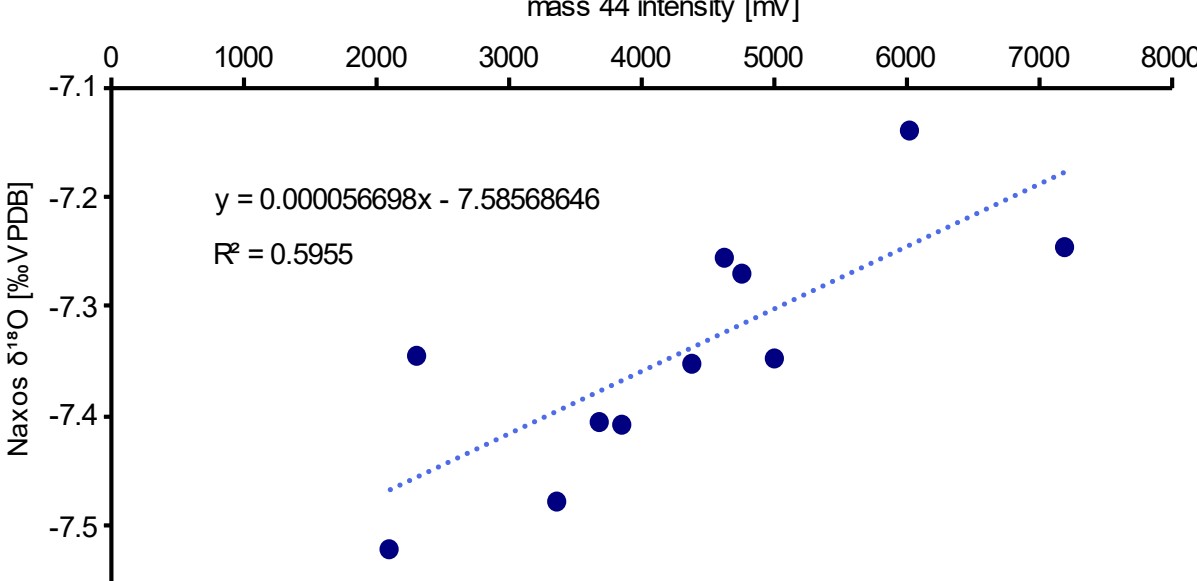

**Figure A2: Linear correlation between the Naxos $\delta^{18}O$ data (11 standards measured) and the mass 44 intensity that was used to correct the $\delta^{18}O$ data.**





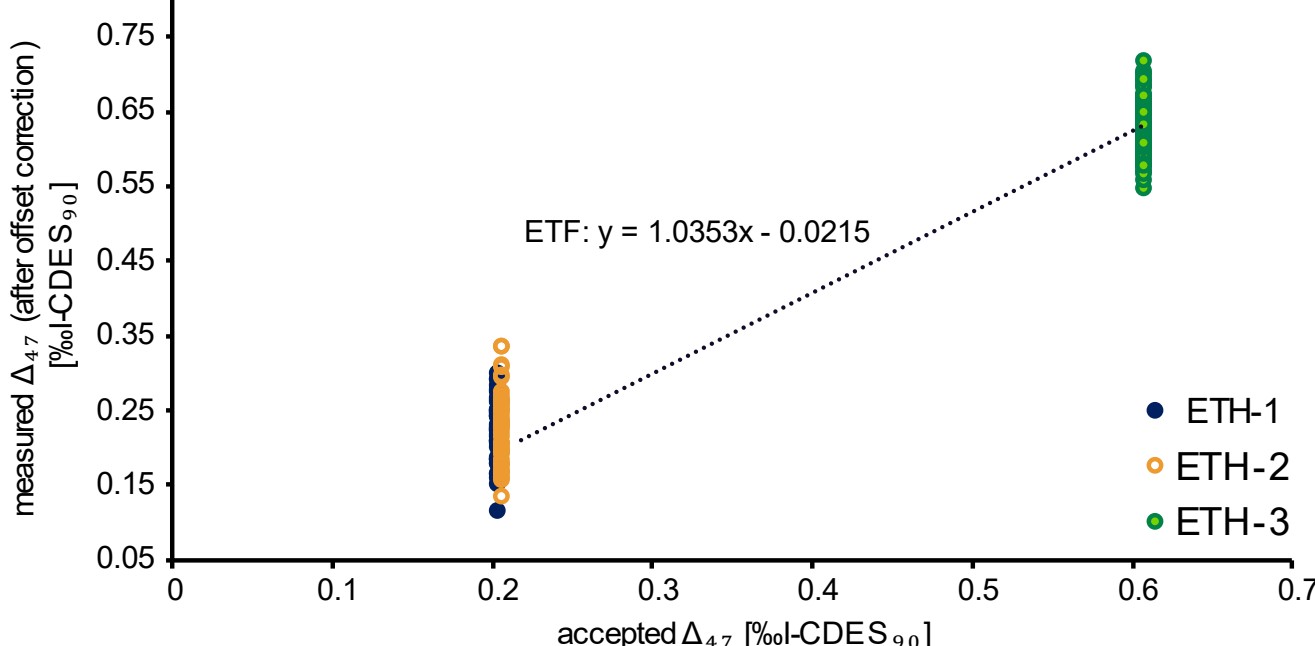

**Figure A3: Empirical transfer function (ETF) constructed from a linear regression between accepted and measured Δ47 values from standards ETH-1, ETH-2, and ETH-3. This function was used to correct all samples and standards and transfer them to the I-CDES90C reference frame.**










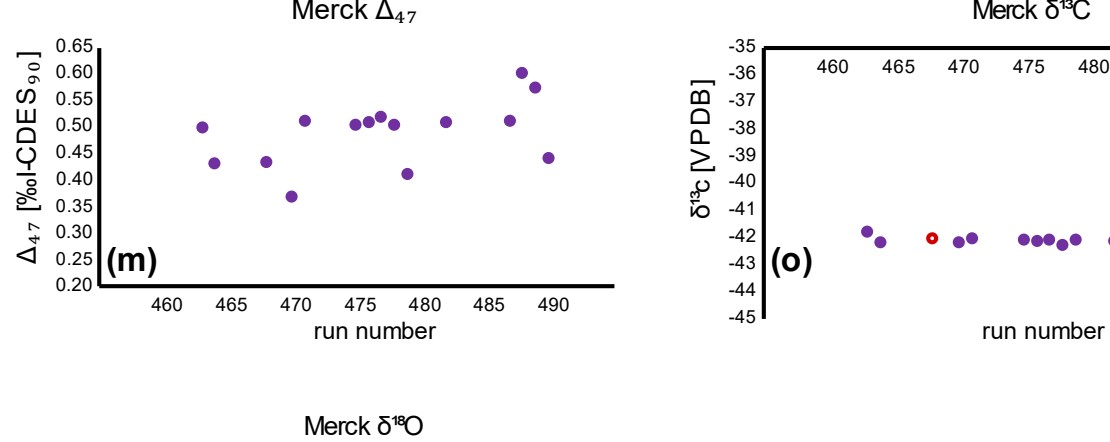

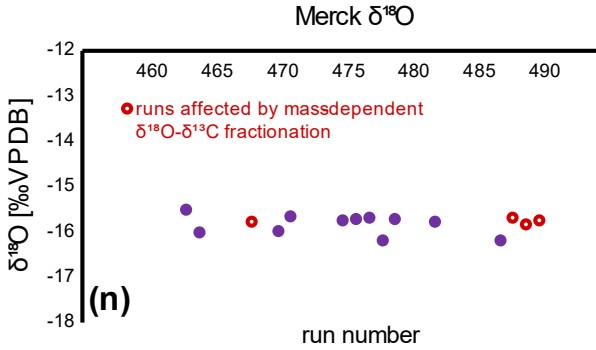

**Figure A4: corrected $\Delta_{47}$ data for external (ETH-3, ETH-2, ETH-1) and internal (IAEA-C2, Merck) standards. (a)-(c): ETH-3 $\Delta_{47}$, $\delta^{18}O$, and $\delta^{13}O$ data. (d)-(f): ETH-2 $\Delta_{47}$, $\delta^{18}O$, and $\delta^{13}O$ data. (g)-(i): ETH-1 $\Delta_{47}$, $\delta^{18}O$, and $\delta^{13}O$ data. (j)-(l): IAEA-C2 $\Delta_{47}$, $\delta^{18}O$, and $\delta^{13}O$ data. (m)-(o): Merck $\Delta_{47}$, $\delta^{18}O$, and $\delta^{13}O$ data. Dates corresponding to the run numbers can be found in the supplementary materials. $\delta^{18}O$ and $\delta^{13}O$ measurements affected by mass-dependent fractionation are marked as red open circles (see sect. 3.5.3).**



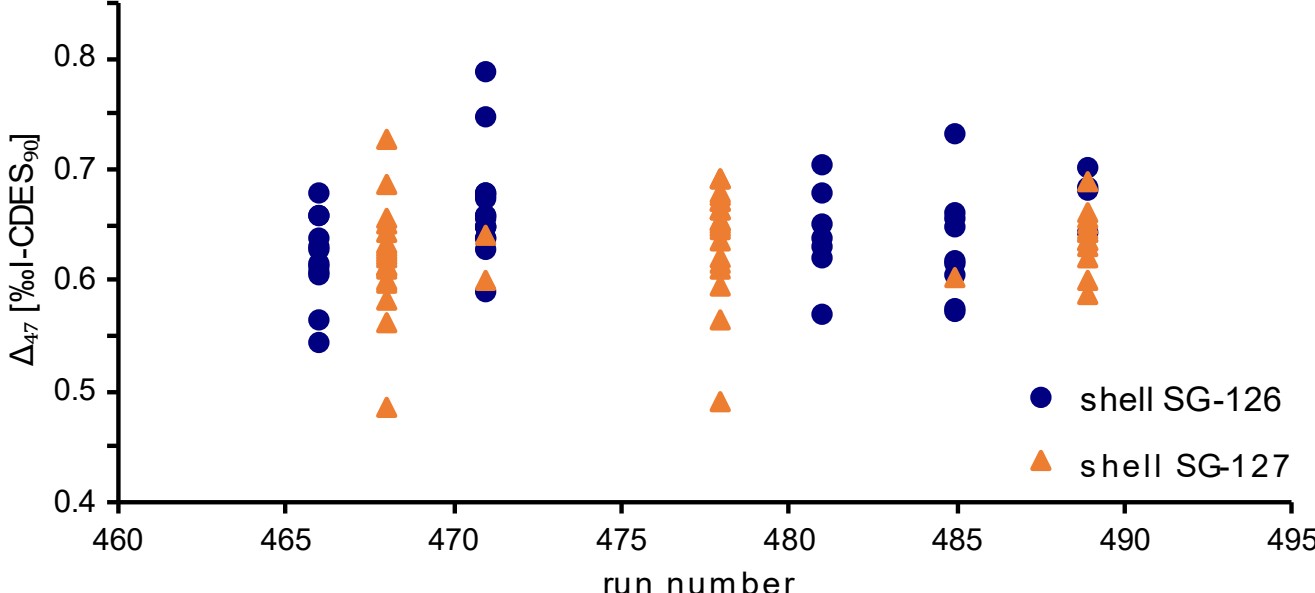

**Figure A5: $\Delta_{47}$ data for specimens SG-126 and SG-127, plotted against the run number as counted in the Utrecht University stable**
**isotope laboratory, to show the (lack of) drifting of values over time. Dates corresponding to the run numbers can be found in the**
**supplementary materials.**




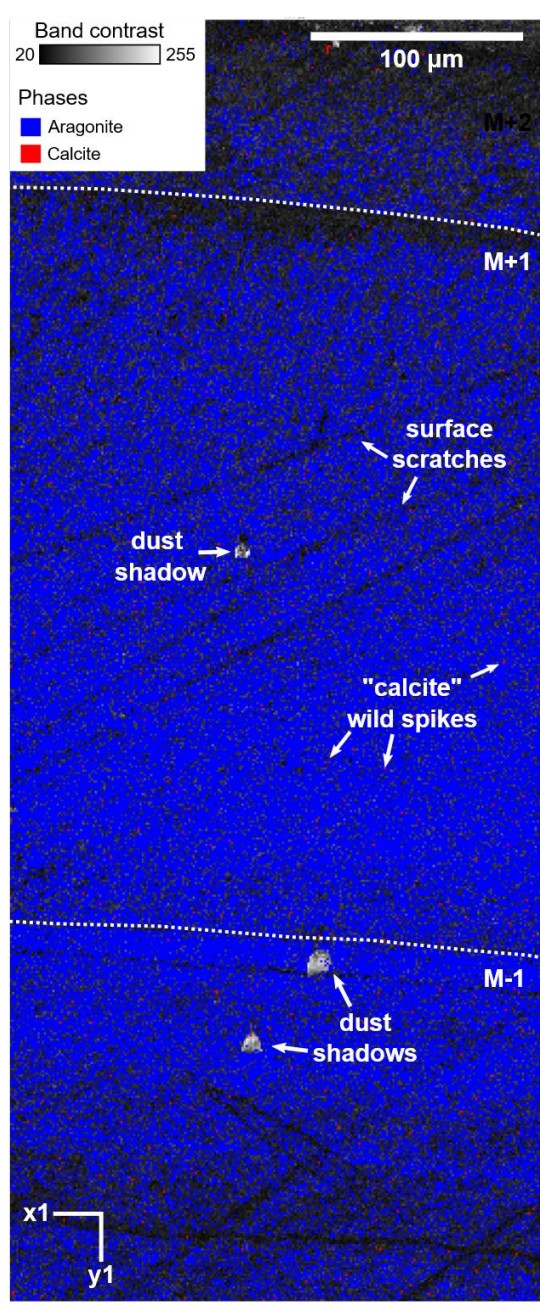

**Figure B1: EBSD map of specimen SG-125 indicating the absence secondary calcite crystals. Map acquisition specifics: 10kV**
**voltage, 1 μm step size. Measured at low vacuum due to excessive charging. Classification: 54.6% aragonite, 45.2% unclassified,**
**0.2% calcite. The map consists of the band contrast overlain by the classified pixels, which are either aragonite (blue) or calcite**
**(red). M+2, M+1, and M-1 correspond to the shell layers as described in sect. 4.1.1. No clean-up was done on this map. The pixels**
**identified as calcite do not show any of the characteristics of actual secondary calcite (large crystals compared to the very fine-**
**grained aragonite, breaking up the aragonitic structure), are always single pixels, and are removed through AZtecCrystal's wild**
**spike removal algorithm. Several dust shadows are also seen in this image; these areas are unclassified. There are also several**
**scratches present on the surface.**



**Code availability**

All code used was written in R. Codes for Monte Carlo error propagation (Benedeni_Monte_Carlo.R) and growth modelling (Benedeni_Growth_Model.R) are available at the Github repository at https://github.com/NMAWichern/Benedeni_benedeni.

**Data availability**

Oxygen isotope, clumped isotope, X-ray fluorescence, X-ray diffraction, and electron backscatter diffraction data are available at the PANGAEA data repository (LINK PENDING; SUBMITTED ON 19/09/2022).

**Author contribution**

NJW designed the study and carried out X-ray diffraction measurements. NMAW carried out stable isotope measurements,
analysed all data and wrote the original manuscript. ALAJ assisted with sclerochronology and ecological and environmental interpretations. SG provided the shell material and carried out the palaeoenvironmental interpretation of the Oorderen Formation. FW carried out the taxonomical analysis of the shells and assisted with ecological interpretation. PK and PC carried out micro-X-ray fluorescence measurements and analysis. MFH carried out electron backscatter diffraction analyses and assisted with their interpretation. MZ supervised the project and aided in data analysis. All authors contributed to
interpreting the compiled data and editing the manuscript.

**Competing interests**

The authors declare that there are no competing interests involved in this research.

**Acknowledgements**

The authors gratefully acknowledge the following people for their help with this work: Leonard Bik (Utrecht University)
made the polished thin sections for microscopy and EBSD work. Desmond Eefting and Arnold van Dijk (Utrecht University) provided technical assistance during isotope analyses. Lucas Lourens (Utrecht University) proof-read the original version of this manuscript. Maarten Zeylmans (Utrecht University) helped produce high-resolution scans of cross sections of the specimens. Pim Kaskes is supported by a Research Foundation Flanders (FWO) PhD Fellowship (11E6621N). Philippe Claeys acknowledges the support of the FWO Hercules program for the purchase of the µXRF instrument and that of the
VUB Strategic Research Program. This work is part of the UNBIAS project funded by a Flemish Research Foundation (FWO; 12ZB220N) post-doctoral fellowship (Niels J. de Winter).





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
