# Peer review of "The fossil bivalve *Angulus benedeni benedeni*: a potential seasonally resolved stable isotope-based climate archive to investigate Pliocene temperatures in the southern North Sea basin"

_EGUsphere, 2022_

## Author Response (AR1)

What is the purpose of this manuscript?

This manuscript deals with the fossil bivalve Angulus benedeni benedeni, looking at its preservation, shell structure, and potential as a paleoclimate proxy. However, I cannot decide which is the authors' main focus – demonstrating high tech preservation assessment techniques, or actually attempting Pliocene paleoclimate reconstruction.

If, as the title suggests, they are assessing whether this species COULD be a good proxy target, they HAVE demonstrated that, for a single fossil collection locality. They have shown that this species grows at a sufficient rate to be able to sample sub-annually for isotopes, and that it is well preserved from this single place/time, meaning it could be used for Pliocene paleoclimate. If this is their question/direction, they should set up the introduction more along the lines of "not all bivalves are good paleoclimate proxies due to growth rates, shell microstructures that are complicated or easily altered, vital effects, etc. Here we investigate whether this new species is going to be useful or not by going heavily into shell structure, preservation, growth rates, and comparing isotopes to previously measured isotopes from more reliable species to look for vital effects." With less focus on the paleoenvironment of these specific samples.

As it is written, there is a big intro on the Pliocene and why the Pliocene is good to study and how paleoclimate data can validate climate models. This makes it seem like the data they are getting in this study will be able to validate models. It is only two shells. And their high-res clumped isotope seasonality didn't work. So it seems like the intro doesn't match the actual deliverables of the study. They don't really have enough paleoclimate data to make a trustworthy paleoclimate story for this place/time. And therefore a lot of the intro material about the geologic setting and age of samples don't matter too much, since in the end you aren't able to say very much about the paleoclimate in that place.

The effort towards clumped seasonality and subannual paleoclimate in general is well-placed. This is an important future direction. For clumped isotope studies in general, and for subannual sampling approaches where you need good shell material over large portions of the shell, preservation is KEY. Therefore, it makes sense to apply many preservation assessment techniques. However, this portion of the paper leaves me again wondering what is the main motivation – is it "are these Pliocene shells well preserved?" or "what is the microstructure of this previously unstudied taxa?". The organization of sections and emphasis on answers to these questions should be considered.

In the end, I think they have the data to back up "these shells are good potential future targets", but should not make too many claims about Pliocene climate. A lot of the bad feeling that comes from "overpromising" what you can deliver could be fixed by a rewrite of the intro with a different angle/focus.

Firstly, the authors want to thank Reviewer #1 for this extensive and detailed review. We are deeply grateful that they took the time to examine and articulate its shortcomings in its current form. The main focus of the manuscript was to highlight that this species can be used as a climate archive, and is found in at least one stratigraphic interval of interest, where it is well-preserved. Or, as the reviewer phrased it, "these shells are potential future targets". The introductory text and discussion on Pliocene Southern North Sea basin climate was merely meant to serve as an illustration of how and where this species may be of use, not as robust paleoclimatic study. The primary goal of the microstructure study is also to

assess preservation, describing the structures is secondary – but still important, in our opinion. However, it is clear that this message was lost due to the extensive discussion of mPWP climate. This indicates an issue with the structuring and wording of the paper on our part. Worse, it indeed results in 'overpromising'.

We therefore agree with Reviewer #1's comments on the general structure of the paper. We propose restructuring the introduction to be more focussed on this species. Consequently, each 'background' section can be condensed into a few lines and incorporated into the introduction, and the background section itself can be removed.

Furthermore, we propose to limit the discussion on our paleotemperature findings to stating that 1) the MAT we found is in line with other proxy and modelling studies for this period and region, and 2) The reconstructed seasonal range, albeit rather uncertain, is in line with other bivalve-based estimates for this period and region.

As we shift the focus to the bivalves themselves, the authors also want to highlight the fact that this study is not only relevant to the specific species *A. benedeni benedeni*. It is also the first study to analyse the genus *Angulus*, and this study forms a basis for further study on related taxa. *Angulus* has a long lineage, going back to the Miocene. Its stratigraphic range covers several important paleoclimatic events beyond the mPWP, such as the MMCO. The relevance of this study is therefore not limited to this singular species.

 We are confident that we can easily restructure and shorten this manuscript to better align with the comments we received.

**Relevant changes made: The abstract, introduction, discussion, and conclusion were rewritten in order to place less focus on the palaeoclimatic implications and the mPWP climate, and more on the potential (and limitations) of the subspecies that is the subject of this study. The background section, except for the geological background, was removed.**

Could these sections be better organized/combined/streamlined?

The methods section is extremely, painfully detailed in certain parts (I guess that's okay if you're fine with an extremely long paper that could serve as a reference for others applying these techniques). This could be helped with moving some of these big detail into the supplement. For example, Table 1. And info like on line 173, listing the grinding elements, and that fact that you are using a front-loader and specifics of the sample holder. Definitely supplement material.

The authors agree that the methodology takes up a large part of the manuscript. Micro-XRF and XRD are rather established methods, and therefore we propose to move part of this text to the supplements (i.e., the information Reviewer #1 mentioned regarding XRD). However, as there is still some level of discussion on best practices within the field of clumped isotopes, it is customary to include detailed descriptions of this method. Therefore, we propose to keep most of this within the main text. Similarly, EBSD is still a relatively new method regarding its application to bivalve shells. Therefore, we deem it useful to also keep most of this text intact within the main text.

**Relevant changes made: parts of the micro-XRF and XRD methodology were moved to the appendix. The other methodology sections were streamlined where possible.**

The results section is painfully long as well. I kept having to flip around to go back to methods, or forward to discussion to see what your findings actually mean. I wonder if you really need results

sections for each of these techniques separately. In particular, 4.1.1 could be wrapped into methods. "we used light microscopy to visually identify 4 layers, previously seen by other authors". OR, perhaps you have a combined results/discussion.

The authors agree that some of the results can be shortened. We do not propose removing entire sections, as we do feel each of these points should be addressed, but we do propose removing superfluous information (such as the 4.1.1 section Reviewer #1 mentioned).

**Relevant changes made: the results were shortened. The results were shortened by ca. 250 words, while the entire main text was shortened by ca. 2500 words.**

I really like the organization of the discussion sections, with headings of diagenesis, shell structure, growth history, paleoclimate archive potential, and paleoclimate findings. If each of these sections began with specific findings and ended with the summary paragraphs currently making up the discussion, that would prevent a lot of flipping back and forth.

We thank the reviewer and propose to include the core findings at the beginning of each of these sections, as we agree that this would improve readability.

**Relevant changes made: we included core findings in the paragraphs were these were still missing,**

Specific comments:

   Abstract is disorganized, following the disorganization of the paper as a whole. The diagenesis results need to come before the isotope findings, for example.

The abstract will be restructured to reflect the content of the revised manuscript, as detailed in the reply to Reviewer #1's general comment.

**Relevant changes made: the abstract was entirely rewritten.**

   Top of page 5: is there a typo here? Two listed temperature anomalies, is one global and the other regional?

Yes, this was a typo. The first 'global' should have been 'regional'. This will be amended.

**Relevant changes made: typo amended.**

   Line 109 – state modern seasonal temp range in the study locality for comparison

In its current form, this would be added to the manuscript. However, after the proposed revisions, the climate of the North Sea area will likely only briefly be discussed in this section. The authors therefore propose to not include these data here, and keep it in the discussion where it is currently stated as well.

**Relevant changes made: this section was removed.**

   Paragraph around line 110 – you state all these findings. Why are they so different? Using different methods with different biases? I need more context here. At the end you mention guesses of d18Ow – is that the core of the disagreement issue?

There is both spatial variability (which can be relevant in such proximal settings), as well as a range in proxy types that have been used to calculate these temperature ranges in the past. On top of that, there is the $\delta^{18}O_{sw}$ issue, but this is of course only relevant for the $\delta^{18}O$ paleothermometer.

However, as the focus of the revised paper will have shifted somewhat away from the paleoclimatic conditions during the mPWP, this entire section will either be removed or significantly shortened. In the latter case, we do propose to elaborate on previously used proxies and potential explanations. But again – none of this is related to questions we can actually answer with our data, so it will likely be removed.

**Relevant changes made: this section was removed.**

Geologic context section seems way too long, given all shells come from a single horizon. Don't need to tell me in detail about the rest of the section, just the time/place/setting of these specific shells. UNLESS you are going to tell me that these species are found throughout and therefore could become useful for high temporal resolution paleoclimate reconstructions. Are there more of A.b.b. in this setting?

We propose to shorten the section to include only relevant horizons, i.e., where the species is found.

**Relevant changes made: the geological context was shortened – but some information had to be kept in as e.g. the sedimentology is important for the interpretation of the water depth, which is important for the temperature interpretation.**

Line 225 – you only have 3 shells to begin with. Why were only 2 sampled for isotopes?

There was no longer enough surface area free at the top of the third specimen (SG-125) to carry out the sampling for isotope analysis. This is due to a section of the shell being powdered for XRD analysis, and part of the shell surface being covered in epoxy as part of the preparation process for light microscopy and EBSD.

We propose to briefly mention this 2 vs 3 shell discrepancy.

**Relevant changes made: this is explained now in section 3.5.1**

Line 228 – you don't want to drill into inner shell layers… but which layers do you mean? May as well name them using your M+ and M- convension. Are you still all within M+2 layer and you mean lower (different time period) growth bands? What is the age of M+1 layer, relative to exterior M+2? I would have said that only the M-1 and M-2 count as "inner shell layer".

This phrasing is indeed confusing. With 'not drilling too deep', we essentially mean 'too far down into M+1'. Considering the thinness of M+2 (ca. 100 micron), drilling into M+1 is inevitable. Drilling somewhat into M+1 does not pose a problem as the layers still represent approximately the same time interval. Drilling down too much will lead to deeper (younger) M+1 material being drilled. Drilling even further will lead to M-1 being reached, which will result in an even larger discrepancy in time of formation.

As it took the authors a paragraph to explain what is meant here, we propose to include a diagram in the supplements explaining how the drilling took place. So as not to confuse readers, we propose to just state that the outer surface of the shell was sampled via drilling.

**Relevant changes made: this has been explained in more detail, and hopefully more clearly, in section 3.5.1. In addition, a figure illustrating the issue was added to the appendix (Fig. B2).**

Paragraph at Line 250 – move this to the intro. In terms of why preservation is so key. Using new techniques like clumped. Here's the background on clumped.

The authors agree with this suggestion. We propose to implement it.

**Relevant changes made: this is now mentioned in the introduction.**

Line 274 – here and other places. Need a better description of replicate vs. average. I know you try and spell that out here, but I was still confused!! Sample = total powder from a single sampling track.

Does that mean following a single growth band, representing a single moment in time. Or do you mean along the growth axis, one tiny bit of powder at each separate drill spot, so small that it = 1 replicate analysis (not really a replicate in that case). Or are you drilling along one growth band to make a "sample", then dividing that into multiple "aliquots".

The authors appreciate Reviewer #1 noting where clarification is needed. The definition of sample vs aliquot is as in the comment's last line. As this is important for interpreting the results, we propose to implement a statement similar to the one below in the main text (in addition to double checking that each term is used correctly in the manuscript):

- One 'sample' represents the total powder that was drilled in one sampling track (which runs parallel to the growth bands). One sample represents a certain amount of time.
- One 'aliquot' represents a small amount of powder from a 'sample' which was then analysed for clumped isotopes. One aliquot = one measurement, but multiple aliquots were taken from a single sample.
- 'Replicates' here are two different aliquots taken from the same sample.
- 'Average' depends on the context.

**Relevant changes made: this is now explained in section 3.5.1.**

Line 281 – ICDES should be ICDES-90

This will be amended.

**Relevant changes made: changed.**

Line 284 – CO2 needs subscript

This will be amended.

**Relevant changes made: changed.**

Line 286 – more details on baseline correction – need a reference?

To correct for negative intensity baseline in the Faraday cups of the mass spectrometer, we performed scans of the shape of the mass spectrometry peaks using reference gas measurements before each run. The baseline on masses 47, 48 and 49 was identified and corrected by regressing the size of the negative intensity outside the peak against intensity of mass 44 (He et al., 2012). We propose to include this information in the revised manuscript.

**Relevant changes made: this is now explained, with the addition of the He et al. reference, in section 3.5.3.**

Line 287 – "method presented in brand" is it a method or do you mean the values provided by brand?

It does concern values, indeed. It will be amended.

**Relevant changes made: this was changed to say values.**

Lie 289 – "drift corrected through bracketing with Eth3" . Is this described anywhere? What type of drift are we talking about? What timescale?

This is a method of correcting for potential short-term (i.e., during one single measurement run, ca. 20 hours) drift. It is common practice for small-volume clumped isotope measurements (e.g., De Winter et al. 2022). We propose to add to the text that it concerns drift during the measurement run.

**Relevant changes made: this is now explained in section 3.5.3.**

Line 291 – after you background correction do you have no slope in d47 vs D47 space for eth1/2? Or are you using the "single step" correction of Daeron?

We find no significant d47-D47 slope (the D47 values for ETH-1 and ETH-2 do not show a significant and consistent difference), and the final ETF correction removes any offset in D47 values related to the isotopic composition (d47) of the aliquot.

**Relevant changes made: none.**

Line 296 – running average …using only standard reps right?

This is indeed a running average through the standards (mostly ETH-3).

**Relevant changes made: it is now explicitly mentioned that this was done using standard reps in section 3.5.3.**

Line 301- If you are leaking gas away enough to mess up your stable isotope values and change the bulk composition, I have a hard time believing that did not change your D47 value. I see the plots, but personally I would not trust any of the data from the red circle runs. I'm confused why it seems to mainly show up in the Eth3 data and not in the others too much. Are you running significantly more ETH3 replicates than the other two?

Yes, we do run significantly more ETH-3 than ETH-1 and EHT-2 during one run, as only the ETH-3 is used for the short-term drift correction (the 'bracketing') and thus has to be measured after every few samples.

While the authors understand the Reviewer's concern, we stand by our explanation that D47 values should not be affected by potential gas leakage. This is supported by theory, by the fact that there is no systematic offset or trend visible within the D47 of the standards, and by the fact that there is no systematic offset or trend visible within the D47 of aliquots from the same bivalve samples.

**Relevant changes made: this argument has been expanded upon in section 3.5.3.**

Line 311 – "from the same aliquots were averaged"…same as what? The groups you just described? I got confused again about what was an aliquot vs. sample. If you were analyzing multiple aliquots per sample, where 1 sample = one drill pathway along one growth band, then were you only including some aliquots from each sample, based on the d18Oc value? Not all aliquots from the same homogenized sample? It is not clear exactly what was averaged here.

The authors regret the confusion. 'Samples' as defined in the earlier reply were grouped based on their d18Oc value (based on either just one d18Oc aliquot, or on the average of all d18Oc aliquots from this sample if multiple were analysed for d18Oc). So, in essence, one sample = one period of time = one (averaged) d18Oc value. Then, ALL D47 aliquots that were taken from this sample were grouped accordingly.

We propose to amend the text to make this distinction clearer.

**Relevant changes made: see previous comment, definitions are given in section 3.5.1**

Paragraph at line 310 – some of this could go to discussion, thinking of trade offs of different thresholds.

The authors agree with this suggestion and propose to move this section to the Discussion.

**Relevant changes made: after restructuring the manuscript, it was decided to keep this in the results after all. No changes made.**

Line 325 – justify use of this paleothermometer. You are not measuring forams. Or discuss how much temp difference choosing a different thermometer would make at the D47 values you are talking about.

We used this paleothermometer as the formation conditions of forams (marine ambient seawater temperature, biologically induced) are the closest to those of bivalves, out of all the calibrations available. Recently, it was shown that the clumped isotope thermometer functions similarly for bivalves and foraminifera (de Winter et al. 2022). However, the clumped isotope paleothermometer does show an offset between calibrations based on these cooler temperature biogenic carbonates and calibrations that include high-temperature abiogenic precipitates.

We propose to include the information stated above in the manuscript.

**Relevant changes made: the above reference, as well as a short explanation, has been added to section 3.6.**

Line 341 – I think you should delete "average D47-based temp". If you are putting d18Osw and max/min d18Oc back into the equation, you will calculate a temp. don't need to input a temp.

This is poor phrasing on our side. Average D47-based temp = the calculated MAT. We do need an input temp to calculate an average d18Osw (assuming that this was constant throughout the year – not necessarily true, but this is discussed in section 5.5). This sentence will be rephrased to explicitly say that the D47-based MAT was used to calculate a d18Osw.

**Relevant changes made: this was rephrased in section 3.6.**

Line 355 – maybe move this to methods so the layer names are already defined.

We agree with this suggestion, as doing so will also make it easier to address the issue related to the mixing of shell layers during drilling that was commented on earlier.

**Relevant changes made: the 'M' naming convention is now explained in the method section 3.1.**

Figure 2 caption – it says layers are marked "1-4". But this is not right.

This was related to an older labelling system and not caught during editing. It will be changed to "M+2 to M-2".

**Relevant changes made: this has been amended.**

Fig 2b – label M+2 layer with a white bar with brackets like in a and c. Also M+1 layer, since I think part of that is showing?

We agree with labelling M+2. However, given the space, M+1 will not be labelled, as it would crowd the figure, and it is already clear from Fig. 2a that M+1 is located below M+2.

**Relevant changes made: these labels have been added.**

Fig 2c – label M-1 layer.

We agree with the suggestion, this will be implemented.

**Relevant changes made: this label has been added.**

Line 380 -384 – this should still be an underestimate right? So mention that.

Yes. This will be clarified.

**Relevant changes made: this has been clarified.**

Fig 3 – what layers are shown here in b? inner (M-1 M-2) and outer (M+1, M+2)? In that case, shouldn't you divide the # of growth increments by two? Shell is laid down both outside and inside myostracum at the same point in time? Show the borders between layers on here.

Only the growth lines in layer M-1 are counted here, which makes up the bulk of the shell close to the hinge. In Fig. 3, a very thin part of M+1 is visible, but no growth lines are counted here. However, to avoid confusion, we propose to add the borders between the layers, as Reviewer #1 suggested.

**Relevant changes made: the borders between layers have been added.**

Lines 397-401 – this section is redundant in many places. Streamline if possible.

The authors agree – this section will be condensed to ca. 1-2 sentences.

**Relevant changes made: this section was shortened.**

Figure 4b – where does the horizontal dashed line come from? Add to caption. Is it the same for all elements?

The dashed line is drawn at 500 ug/g, and indicates that almost all measured datapoints have Fe and Mn values below that. There should have been a label at this line, but the authors feel that it is rather redundant in any case.

We propose to remove this line to avoid confusion.

**Relevant changes made: the line was removed.**

Section 4.3.2 – I am not sure why any of this matters? Nowhere is it connected to something meaningful. Are you hoping others would use this to test preservation? Or does this matter for physiology? Does it compare similarly to other taxa? Why is this worth your time to measure?

Reviewer #1 is right in that we do not discuss these data later on. These data were included as it is a routine part of EBSD studies, and it may be relevant comparison material for people who are studying other (tellinid) bivalves with EBSD. There is currently limited information available on bivalve EBSD data compared to methods like XRF, and we therefore think it is worthwhile to add to this body of information.

**Relevant changes made: none.**

Figure 5 – as with comment above. I am not sure what to take away from this figure. I don't think it needs to be in the main paper.

See comment above. It may be relevant for others working on bivalves and using EBSD, even though it is not the main focus of the paper.

**Relevant changes made: none.**

Line 463 – Is this internal reproducibility for one replicate or the average of N replicates? Describe more clearly.

This is instrument error on the samples – reproducibility on the replicates is discussed in the next paragraph. This will be clarified, and the number of measurements each estimate is based on is added to the text for transparency.

**Relevant changes made: this has been clarified, and moved to section 3.5.2.**

Figure 6 – not clear what I should take away from this either. Which shell is being shown? Add to caption.

It is specimen SG-127, this will be added to the caption. This figure is shown to 1) indicate that the original structures still exist, confirming good preservation, and 2) which microstructures these actually are. The latter is of interest to other workers that study bivalves using EBSD, since as of now there are still relatively few studies that use this technique on bivalves. We believe it therefore to be useful to provide this "example" material and add to the growing, but still limited, body of work on bivalve EBSD analysis.

**Relevant changes made: it was added to the caption that this concerns specimen SG-127. No other changes were made.**

Figure 7 – For all the detail provided on the preservation techniques, there is a real lack of detail, or lack of transparency about the averaging of isotope data and how it was sampled. Show your raw data in a supplementary figure.

We respectfully disagree that there is a lack of transparency about the sampling and averaging of isotope data. All data processing steps are described in detail, and all raw data are available in the supplements.

We propose to include the raw data plot in the supplements, as the reviewer suggested. In addition, we remain open to suggestions to further improve the transparency of our isotope data processing.

**Relevant changes made: the raw data plot was added to the appendix (Fig. B6).**

Line 485-487 – I am confused about this uncertainty calculation. Why is the number so much bigger than reps of Naxos over a long period of time. Is there really that much heterogeneity in aliquots of the same "sample"? How were the aliquots drilled. Only here in the results does it say they were not homogenized before being analyzed. That should come in the methods section, when describing sample prep and drilling. Section 3.5.1 would be a good place for you to define sample, aliquot, etc.

The authors agree that it should have been stated that the samples were not homogenized before analysing in the methods section. This will be amended.

As Reviewer #1 pointed out, the replication error on the Naxos is much smaller. Thus, the authors infer that the larger error on multiple aliquots of the same sample arises from the heterogeneity of the sample.

**Relevant changes made: it was added that the samples were not homogenised in section 3.5.1.**

Line 489-this section until line 496 could move to methods section, if you're separating out all the methods. Or if not, then combine like this for other sections, where you describe the calculation then results right after.

The authors agree that this section fits better in the methods section and will be placed there.

**Relevant changes made: this section was moved to the methods (3.7).**

Line 517-518 – yes they could be determined, but the error bars are big. The seasonality is apparently very low. You should state the absolute #s, the temps that large symbols in Fig 9 correspond to. Actually, present the mean, error bar, and mean temp, error on temp on the figure right under the other text about the threshold.

The authors disagree that 'error bars being large' must correspond to a low seasonality. The clumped isotope seasonality estimation results in an underestimation of the actual seasonality, for reasons stated in section 5.5.

As we understand it, the reviewer suggests to alter Fig. 9 to have D47-calculated temperature instead of D47 values on the y axis. The reason that this was not presented with temperatures is because we ultimately do not use temperatures calculated in this way, because of their large errors. As we feel that

it would be confusing or even misleading to plot temperatures that are not that meaningful on this plot, we propose to keep it as it is.

**Relevant changes made: this issue has been stressed in section 4.4.4.**

Line 521 – when you say "more measurements are required" be specific that it's not just more measurements, its more measurements at d18Oc> 2.4, at the seasonal extremes. How many reps would you need to believe this? For a single homogenous sample, how many reps do you need to get +/- 2 degrees C error on that sample (near earth surface temps)? How does that compare to what you have?

The text will be amended to state that it is indeed these measurements corresponding to more extreme d18Oc values that are needed.

The authors admit that narrowing the temperature down to +/- 2 C at 95%CL will be a difficult task that will likely require several hundreds of measurements (considering the MAT calculated here has +/- 3.8 C with N=103). Confidence level scales with 1/sqrt(N), so progressively more aliquots would be needed to bring the error down. However, obtaining ca. 100 measurements at >2.4 d18Oc should certainly be doable when combining data of 2-4 complete shells (as shown in A1, specimen SG-127 could not be sampled fully, limiting the data from the 2 shells used here).

As the focus of the manuscript will be shifted towards this species as a climate archive, we propose to discuss this in the discussion section.

However, we must add that large errors in general are an inherent limitation of the clumped methods, and the aim of the manuscript is not to discuss the pros and cons of this method in general. It is merely used as a tool here. The discussion will therefore be limited.

**Relevant changes made: an estimate of 3-5 shells (to be conservative) was added to the discussion in section 5.4.**

Line 525 – I don't understand this sentence. "so they better represent seasonal extremes". Is this discussing data you have, or hypothetical additional sampling you might do?

This is discussing hypothetical additional sampling. As the wording is confusing, it will be altered to 'would better represent seasonal extremes' to reflect that this isn't something that has currently been done here.

**Relevant changes made: wording has been changed as described above.**

Line 530 – these last few lines of the results section seem to come out of nowhere. Why mention standard deviation of single samples here. That is not what is being plotted anywhere – always averages.

This paragraph (starting from 'The external reproducibility…') is indeed unfortunately placed. It relates to the general performance of the clumped data, not to the samples or temperature interpretation. This, and other lines concerning instrument performance, will either be moved to the methodology section, or to the top of the results.

**Relevant changes made: this section was placed in the methods section (3.5.3).**

Figure 10a – state the MART as well. Maximum, minimum and average MART given errors on summer and winter Ts. How does MART from D47 compare? Only slightly less?

As explained in section 4.4.3, we do not consider the D47 seasonal temperatures reliable enough to plot and discuss.

**Relevant changes made: none.**

Figure 10b – plot the D47-based summer/winter estimates for the 20/21 thresholds, including error bars. This will highlight how big the error in temp is from these results. To me it looks like the summer and winter estimates are both within error of the d18O estimates, and the summer average is actually very close. The D47 average would be within the smaller d18Oc based error bar. How did you choose 3 d18Oc data points to average together? Seems very small. Why not average those >2.4 and< 0.9? Or why not just take the absolute max/min d18Oc value? Justify this. Or report multiple of these and compare.

Based on several comments, the authors think that we have failed to stress that we do not actually calculate, or want to show, summer and winter temperatures based on D47 only, because these have too high error bars to be reliable. This is mentioned in section 4.4.3., but based on the confusion, this should be stressed more, and we propose to do so in the revised manuscript.

Similarly, as is mentioned in section 4.4.3, the values of 2.4 and 0.9 d18O are merely selected to *illustrate* the issue of enough clumped datapoints vs better capturing the actual seasonal range. They are arbitrary cut-off values.

Finally, the 3 lowest and highest d18O values were chosen to calculate the potential *maximum* seasonality that was recorded, not the average seasonality throughout the specimens' lifespan. The single highest datapoint could have been chosen as well, but an n of 3 was chosen to avoid relying on one single 'peak' of the d18Oc record.

We propose justifying the use of 3 datapoints in the text, as was explained above.

**Relevant changes made: the use of 3 datapoints was expanded upon in section 4.4.4.**

Fig 10 overall – the smaller MART from D47 suggests the d18Oc thresholds were too close together. I wonder how the d18O-based seasonality would compare to the D47 seasonality if you averaged all points with the same thresholds. It would reduce the d18O-based seasonality from what you plotted in 10b.

See our answer to the previous comment. There is no D47 calculated seasonal temperature, because these data were deemed unreliable. The reviewer's last sentence - 'It would reduce the d18O-based seasonality from what you plotted in 10b' – is precisely the reason why we picked the extremes. We *want* to figure out what the maximum potential recorded seasonal range was. There are already unavoidable physiological and methodological factors that result in an underestimation of this range (see section 5.5), even if one does not use the clumped isotopes (which we do not use, to stress this again). It therefore makes no sense to us to then also average d18Oc values, when the error bars on these measurements are relatively small.

**Relevant changes made: none.**

Linw 595-605 or so – this description of the layers needs to come a lot sooner.

This section was placed here as 'assigning' the structures to existing ones found in other species was seen as technically being comparison with literature and thus more suitable to the discussion. However, the authors can see how this may be confusing for the reader.

We propose to move this to the results section. The results themselves can then be shortened, to avoid repetition.

**Relevant changes made: this section was moved to the results (4.3.3).**

Line 624 – show the d13C profiles somewhere? Supplement at least.

The d13C data are included in the supplementary data. As it is not relevant to the study itself, we propose keeping it there as data only, not as a supplementary figure.

**Relevant changes made: none.**

Line 655 – you have this backwards. Limited sampling resolution and missing growth lines means your estimates are LOWER than reality.

This is true, it should have read '..more likely to be under- rather than overestimated..', rather than '..more likely to be under- rather than overestimated..". This will be amended.

**Relevant changes made: edited as explained above.**

Line 663 – how would you expect the spring/neap tides to show up? In growth band thickness? Prove that it's not there… how would you expect to ID it?

As mentioned in the text, we would expect these to show up as asymmetric bundling, either in growth band colour or thickness. This is not what we observe here.

Our (very crude) growth line counting showed a rough fit with a growth line being formed every 29 days, this is unlikely to represent the lunar cycle as this would imply an asymmetry between the first and second spring-tide cycle, which is rare. If the growth lines corresponded to spring-neap tide, we would expect bimonthly lines, not monthly. Hence, either the 'monthly' cycle of 29 days actually represents an internal rhythm, or our estimates are off by a very large margin regarding growth line counting.

For a more robust constraint on growth lines, we need more specimens, more sophisticated techniques (e.g., etching the specimens), and the application of statistical methods such as spectral and frequency analysis. However, this is beyond the scope of this study.

**Relevant changes made: this issue is further explained in section 5.3, and some of the discussion was removed as it was too speculative.**

Line 673 – "a larger dataset"… how much larger?

This is indeed an important estimate to include. We estimate 2 (full) to 4 specimens to obtain enough clumped measurements that correspond to high and low d18Oc values. We will include an estimate in the revised manuscript.

**Relevant changes made: see previous comment  - an estimate was added to section 5.4.**

Line 690 – modeled average GLOBAL sst

'global' will be inserted in this sentence.

**Relevant changes made: change proposed above was implemented.**

Line 703-704 – d18Oc suggest higher seasonality than modern. But are you actually showing interannual variability instead of annual variability. By averaging only the highest and lowest THREE points you are capturing the warmest summers and coolest winters, not the average summer/winter. Expand your thresholds and you will get closer to the modern MART. Also it seems like D47 seasonality would be closer to modern MART.

See previous replies regarding D47 seasonality and maximum seasonal range.

**Relevant changes made: none.**

Line 722 – if a strong positive correlation between T and d18ow is there today, how would that affect your d18Oc seasonality? It should enhance it…So perhaps the D47 reduced seasonaility is plausible after all. Do you see correlation between d18Osw and D47-temp? make a figure here.

As is mentioned in the text, a positive correlation between T and d18Osw will actually dampen the d18Oc signal, therefore making a larger-than-estimated seasonal range more plausible. High

temperatures during summer will drive d18Oc to lower values, but this is dampened by the fact that the d18Osw is relatively high due to evaporation, resulting in overall less low d18Oc values in summer. The reverse is true for winter. See also Ullmann et al. 2010 and references therein. Their data are also from the southeastern North Sea, and thus applicable to our study.

See previous replies regarding D47 seasonality.

**Relevant changes made: the issue is explained in more detail, including the above reference, in section 5.5.**

Supplement

Figure A2 – why is this correction needed? How much range in M44 intensity is there for sample reps? Are you weighing out each aliquot? Shouldn't they all be fairly consistent? Can you show the same relationship for other standards? It should be present in all of them right? What voltage did you correct to using this equation?

Each aliquot was weighed out, but the recommended range for each aliquot as per the laboratory's protocol is quite large (50-100 ug, as mentioned in the methods section. These values can also be found in the supplementary data). This corresponds to a rather large range in M44 as well, which is why the correction was carried out.

There was one other carbonate standard present, with N=4. The slope for this is very similar, but the intercept differs. This is expected of a correlation plot with such a low N. The d18O values were corrected based on their M44 value and their raw d18O value.

**Relevant changes made: it is explained in the figure caption that the correction is necessary due to the large weight range that was used. Note that this is now Figure B3.**

Figure A3 – "after offset correction"…what does this mean? Clarify. Shouldn't this plot be reversed with the accepted value on the y axis? Is this all the data from one measurement session or from the whole time samples were run? Were all samples corrected in the same measurement window? What is the benefit of showing this plot?

"Offset correction" here refers to the "bracketing" practise as explained in a previous comment. The authors agree that this should be clarified; we propose adding it either to the plot itself, or to the caption.

This is from the entire time samples were run. We propose clarifying this by mentioning the measuring timeframe in the caption. All samples were corrected in the same measurement window. We propose mentioning this in the caption for clarification.

The plot is shown for transparency reasons, as it is common in the clumped community to give a detailed methodology.

**Relevant changes made: The axes are reversed in the new plot, and it is explained in the caption during which timeframe the standards were measured. Note that this is now Fig. B4.**

Figure A4 – different scales on the D47 plots make the scatter look a lot different. Are these points replicate level? Aliquot level? Sample average of multiple aliquots? How much powder/integration time for each of these points?

These are all individual aliquot measurements (which is perhaps obscured by the fact that many points overlap, making it seem like a lower number of measurements). We propose clarifying this by mentioning it in the caption.

The general range of powder for an aliquot can be found in the methods section. Data on powder and integration time for each individual measurement can be found in the supplementary data. We do not propose to make alterations regarding this comment.

**Relevant changes made: it is mentioned in the figure caption that these are single aliquot datapoints. Note that this is now Fig. B5.**

Figure A5 – this does not show me drift over time since I don't know which samples each of these points come from, which are summer vs. winter? Show this for Merck or IAEAC2 isntead? Drift in sample values could be real signal.

These are all measured samples. For the purpose of this plot, it is not relevant to know whether these are summer or winter datapoints, as it merely illustrates that there is no visible drift over the several weeks that the samples were measured. The data for Merck and IAEA-C2 are shown in A4. Any real signal is still preserved, as we did not correct for drift (since we didn't observe any).

The plot also illustrates the range in D47 values for the samples.

However, as this information can be learned from the supplementary data as well, and is not that pertinent to the paper itself, we propose removing this plot.

**Relevant changes made: the figure in question was removed.**

**Paul Butler – referee #2**

This paper uses oxygen isotope and carbonate clumped isotope measurements in the shell of the extinct tellinid bivalve Angulus benedeni benedeni to assess the potential of this archive as a temperature proxy for the mid-Piacenzian warm period (ca 3 Ma) in the southern North Sea basin. A mean annual temperature of 13.5±3.8°C is determined, which is 2.5°C warmer than today and 3.5°C warmer than the pre-industrial North Sea, consistent with global Pliocene temperature estimates of +2-4°C compared to the pre-industrial climate. Seasonal temperatures could not be reconstructed because of a limited amount of clumped isotope data, but the growth band width apparently allows for these to be determined in future research.

This paper also discusses shell microstructure in great detail using a number of different methods.

In summary: "The bivalve A. benedeni benedeni is suitable for high resolution isotope-based paleoclimatic reconstruction and it can be used to unravel the marine conditions in

the Pliocene southern North Sea basin at a seasonal scale, yielding enhanced insights into imminent western European climate conditions."

The authors thank Paul Butler very much for his extensive review of our manuscript. We do not propose to make large structural changes based on his comments. However, we will address the several specific comments below.

**Relevant changes made: none.**

L 95-96 "Proxy data indicate strong warming in the North Atlantic region during the mid-Piacenzian, with global mean sea-surface temperature (SST) anomalies of up to +4-7°C instead of the global +2-4°C" Presumably that first use of the word "global" should be "regional".

This is correct, the first 'global' will be changed to 'regional'.

**Relevant changes made: this was corrected.**

L 104 "Insights into" rather than "Insights in".

This will be corrected.

 **Relevant changes made: this was corrected.**

LL 110-115 There is quite a wide range of estimates of temperatures and seasonality for the SNSB, and by implication this is attributed to different assessments of the ocean oxygen isotopic composition, although I guess it could also vary with site, proximity to the coast and perhaps other variables. Notably, the description of the site environment in lines 137-142 suggests a potentially wide range of marine oxygen isotope values at that site alone. That's one reason why it would be nice to have a map showing the original context in some detail.

There is both a range in data collection site and with that, proximity to the coast, as well as a range in proxy types that have been used to calculate these temperature ranges in the past. With this in mind, a map/figure illustrating these ranges would indeed be clearer than just text.

However, after reviewing the focus of the paper based on insightful feedback, we have decided to shift the focus somewhat away from the paleoclimatic conditions in the SNSB. After restructuring the paper, a figure as described above would no longer be relevant to the main study. We have therefore decided to not add it.

**Relevant changes made: the section in question was removed.**

L 227 "… not to drill too deep into the inner shell layers …"

Indeed a better way to phrase this, it will be amended.

**Relevant changes made: see previous comment to referee 1; it has been clarified and a figure has been added to the appendix.**

L 380 "fewer growth lines"

We are unsure which part of the phrase this comment refers to.

 **Relevant changes made: none.**

The technical descriptions are extremely detailed, and I am not sufficiently expert in the technicalities to easily identify anything amiss with them. However, given the background of the authors, they clearly come from a position of expertise. They are also very well written.

The descriptions are perhaps too long-winded for the main text. We intend to shift some sections of them (e.g., XRD instrument settings, some of the micro-XRF details) to the supplements of the paper, so that they are still accessible, but do not occlude the main text.

 **Relevant changes made: see previous comment to referee 1. Parts of the uXRF and XRD methodology were moved to the appendix.**

The conclusion appears to show consistency with previous estimates of temperatures during the mid-Piacenzian in the southern North Sea basin. It is indicated that growth increments are monthly (probably corresponding to the monthly synodic lunar-tidal cycle) and that the animal lived about 10 years.

This is correct. Our main finding is that the studied species can be used as a climate archive.

**Relevant changes made: none.**

I think this paper provides a solid template for further work with clumped isotopes. It's been a difficult technique to develop for bivalves, and there clearly remain challenges in terms of the amount of material and number of samples needed to overcome the analytical error. But this paper will help future researchers to make realistic determinations of how many samples they need to process.

We thank the reviewer for this remark and are glad to read that he thinks our manuscript has merit as a template for future clumped isotope studies in molluscs.

**Relevant changes made: none.**

Figure 1 shows the location of the shells. It would also be interesting to see the topographical configuration in the area at the time the shells were living and the context in which the shells were living

Considering the fact that the English Channel was closed at the time these bivalves lived, this may indeed be an important addition that can be quickly added. We thank Paul Butler for the suggestion. We will clarify the difference in topographical and oceanographical conditions between the mPWP and modern.

**Relevant changes made: the figure was not amended.**

Figure 2 shows the different shell layers in detail

We thank the reviewer and are glad to have confirmation that the figure is clear.

**Relevant changes made: none.**

Figure 3 shows the counted growth lines although it isn't always easy to see the growth lines on this image

The authors agree that it is indeed not the highest resolution image to discern growth lines. Unfortunately, we did not produce higher resolution images. We therefore intent to keep this image as it is.

**Relevant changes made: none.**

Figure 4 shows the aragonitic composition of the shell

No comment.

**Relevant changes made: none.**

Figure 5 is a very detailed EBSD map

No comment.

**Relevant changes made: none.**

Figure 6 shows the shell structures identified by EBSD

No comment.

 **Relevant changes made: none.**

Figure 7 shows the del18O annual cyclicity with the changing shading of the growth bands

No comment.

**Relevant changes made: none.**

Figure 8 also shows the annual cyclicity

It also shows the fitted growth curves – albeit admittedly based on few datapoints.

 **Relevant changes made: none.**

Figure 9 good illustration of why the clumped isotope measurements cannot distinguish summer and winter temperatures.

No comment.

**Relevant changes made: none.**

**Anonymous referee #3**

Review of the manuscript: The fossil bivalve Angulus benedeni benedeni: a potential seasonally resolved stable isotope-based climate archive to investigate Pliocene temperatures in the southern North Sea basin, by Nina Wichern and co authors

The focus of the paper concerns the reconstruction of the seasonal paleotemperature gradient during the Pliocene in the North Sea using the bivalve species Angulus benedeni benedeni.

To do so, the authors use a multi tool approach, using several proxies to test the preservation of the samples, and also sclerochronology and stable isotopes measurements (d18O and â^†47).

The main conclusion is that the mean annual temperature was 2.5 °C warmer than today in the North Sea. This result and this kind of approach is very important to better understand the evolution of the temperature in the next decades and the consequences of such a strong warming. However, most of the paper focused instead on a methodological approach dedicated to test the use of this bivalve species as a paleoclimatic archive. Specifically, the authors test numerous methods to verify the proper preservation of samples.

This approach is entirely commendable and extremely important, but is probably too extensive for such recent samples and is out of context with the problem exposed in the introduction.

It is therefore necessary either to shorten this part or to modify the problematic and the title of the article on methodological questions.

Given the distribution of data and approaches tested, I recommend that the authors refocus the issue on the question of using bivalves as a paleoclimatic archive, with an example of application to the Pliocene of the North Sea. Indeed, the authors have only two shells, which present different geochemical profiles.

Moreover, as the authors themselves point out very well, there is an uncertainty concerning the estimation of paleotemperatures, due to several points, such as the value of d18Ow, the resolution of the sampling or the variations of the growth rate of the fossils.

It should be noted, however, that the approach developed by the authors is extremely promising in coupling several tools to both reconstruct temperature variations and above all to understand what the analyzed bivalves are able to record in terms of environmental signal and therefore what it is possible to restore

Given all these points, I recommend the publication of the article after a reformulation of the problem, more in line with the results and the tools used.

The authors thank Reviewer #3 for their extensive and careful review. Their concerns are in line with some the authors' own concerns upon submitting this manuscript. Our main goal was to show that this species is a promising climate archive, occurs in relevant stratigraphic intervals, and can be analysed with a range of modern methods. The application to North Sea paleoclimate was largely meant as an illustration of this potential. After reading the reviews, this was clearly not interpreted as the main conclusion. This reflects a structuring and wording issue on the author's side. We therefore agree that the title, introduction/background, and discussion should be shortened and reformulated to reflect this goal. To this end, we propose to reformulate the introduction focussing on this species and its potential instead of the mPWP North Sea climate. The main points of the sections in 'background' will be incorporated into this introduction, and the rest will be removed as it is too extensive and distorts the proposed goal of the study. Similarly, we agree that the discussion contains too much detail about the paleoclimatic interpretations and extends beyond the illustration that it was meant to be. We therefore propose to keep the paleotemperature discussion along the lines of 'the MAT we found is in line with other proxy and modelling studies for this period and region. The reconstructed seasonal range, albeit rather uncertain, is in line with other bivalve-based estimates for this period and region', and go no further than that. As has been pointed out, the rest is too speculative.

A final note on the necessity of several diagenetic screening techniques: we want to point out that it is possible for relatively recent specimens to be significantly altered to the point of no longer being useful for isotopic studies. An example would be the complete loss of aragonitic shells in the slightly older (Zanclean) Sudbourne Member of the Coralline Crag Formation in eastern England (Balson et al. 1993).

**Relevant changes made: see the first reply to referee 1. The introduction and related sections have been rewritten to better reflect the outcome of the study.**

Apart from this general point, here are some other points of questioning:

2.3. Geological context

I don't understand the justification of the paleobathymetry of the bivalve layer. Why do the authors choose a depth of 20 m for the level when the bivalves indicate a depth of 35-45 m? What is the argument for deciding between the different signals? Tt is crucial for the interpretation of temperatures afterwards.

The sedimentology suggests deposition above wave base, which is generally closer to 20 meters. While, for transparency, we included the bivalve-based depth estimate, we interpret the sedimentology as a more reliable indicator of depth as it is based on physical processes instead of (more opaque) bivalve paleoecology. Moreover, the fact that it is above wave base is relevant for the interpretation of the reconstructed temperature signal, as it makes it more likely that this individual recorded SST-like conditions instead of sub-SST conditions.

We propose to amend the text so that this decision to go with the 20 m estimate is clearer.

**Relevant changes made: additional references and arguments have been added to section 2.**

Sclerochronology

The authors state that it is not certain that all increments have been identified.

Why didn't the authors try to use finer observation methods than the only natural light observation, like cathodoluminensce or Mutvei staining, which make the identification of increments easier?

We thank Reviewer #3 for this comment, it should indeed be addressed why no methods better suited for this purpose were employed. The authors did not use finer observation methods such as the ones mentioned because this was a) beyond the scope of this study, time wise, and b) the authors' main concern regarding growth rate was assessing whether this species grew fast enough to enable the sampling of multiple years at high resolution. This was already confirmed by the $\delta^{18}O_c$ record. Determining the details of growth rate and evolution of this species was of secondary importance in this study. It could, however, form a valuable part of a follow-up study with more specimens. The authors therefore do not propose to include this in the manuscript, but we do propose mentioning that a more detailed sclerochronology work is a good avenue for further study.

**Relevant changes made: we added a paragraph concerning this topic to section 5.4.**

Regarding the identification of bivalve growth rates, did the authors attempt to measure the width between two successive increments and do a frequency analysis of the counts?

We thank Reviewer #3 again for this comment on a methodological limitation that should be addressed. We propose to highlight this limitation of our methodology and propose more detailed investigation of the sclerochronology of this bivalve taxon as an avenue for future study. As it stands, our description of the growth increments are sufficient to meet the main aim of our study, namely to show that the taxon can be used as a climate archive.

**Relevant changes made: see comment above.**

Also, how did the authors take into account probable winter growth stops in the identification of increment mineralization frequencies?

The authors did take probable winter growth stops into account in a qualitative way for the interpretation of the temperature reconstruction (by correlating the darker bands on the surface of the shell to high $\delta^{18}O_c$ values). However, the reviewer right in that it is missing in the discussion on the growth increments.

If a winter growth stop occurred, the duration of deposition of the counted growth increments is in fact shorter than we expected. However, since we do not have enough evidence to quantify the duration of a potential growth stop, we cannot reliably estimate how much shorter-lived our growth increments likely were. We propose to include this discussion in the revised manuscript.

**Relevant changes made: this has been expanded on in section 5.4.**

In relation to this point, I do not understand what is the conclusion of the authors on the frequency of mineralization of shells and especially what are the arguments chosen to arrive there?

Based on the shape of our d18O curve and the counting of growth increments in the cross sections through the shells, we estimated the duration of an individual growth increment. Since we could not constrain the duration of winter growth stops, we did not include this in our estimate. We acknowledge that the resulting estimate is flawed, and we propose to acknowledge this in the revised manuscript as well. As a result, our estimated durations of the growth increments are maximum estimates and we will specifically mention that the duration may be shorter in case a winter stop occurred. We consider it unlikely that a significant part of the winter season is missing based on the seasonal range in d18O and temperature we reconstruct, but the reviewer is right in that we cannot exclude it based on our evidence.

**Relevant changes made: this has been – hopefully – clarified in section 5.4.**

---

## Author Response (AR2)

**Dear Dr. Wichern and Co-Workers,**

the reviewer was pleased with the overall revision of the manuscript but suggested a few more minor adaptations, which I believe will be easy for you to implement. It seems the reviewer made a technical mistake and entered their suggestions in a form that is only visible to the editor. I am therefore pasting the very helpful comments here into my response to you. Please be invited to upload a new version of your manuscripts after minor revisions. Please provide a manuscript version with changes tracked along with your submission. Let me know in case you have any questions. Best

Tina Treude

Reviewer #1 Comments:

Since the first version, the reframing of the introduction makes the scope and purpose of this paper much clearer. Overall the paper is much easier to read than before. I applaud the authors revisions and encourage acceptance following minor/technical revisions detailed below. All these suggestions should be able to be addressed by adding a single sentence or few words.

The authors are glad to hear that the revisions have improved the clarity and readability of the manuscript. We want to thank the reviewer for their helpful comments, and we have incorporated these further points of improvement into the manuscript.

Line 64: "predicted for moderate IPCC pathways".... We are already > 400ppm? Is this a relevant mention anymore?

It is true that this is no longer relevant for the atmospheric  $CO_2$  concentrations, but it remains relevant for (short-term, end-of-century) temperature changes. We have amended the text to reflect that this comparison focusses on temperatures only.

Figure 1 : the glauconitic sand and clayey sand are hard to distinguish. Is there any area that is NOT clayey? Make this visually more distinguishable?

Most intervals have some clay content, except for the Luchtbal Member. The top of the Oorderen and the entire Kruisschans Member are more clay-rich than the rest of the Lillo Formation. The horizontal lines that indicate clay content have been made longer and thicker. Unfortunately we had to revise some of the stratigraphy at the last moment, so the figure has been altered in general.

Line 120: "Their" should be "they are"

Thank you, this has been corrected.

Line 171 + following paragraph: how much material (weight, mg or ug) is typical for one drilling transect (one sample)?

This information is already noted at the beginning of section 3.5.1. Line 162: 'Samples of approximately 100-300  $\mu$ g were taken from the outer surface of two specimens..'.

Line 216 – this bracketing correction for ETH3... Is this supposed to be equivalent to the ETH1/2 slope correction step? I can see how that is partially getting at the same issue – you are basically making a line through two points that are (d47 of measured ETH3, D47 of measured ETH3) and (d47 of 0, true value of ETH3) to access that slope, except it does not take into consideration the d47 of your samples. If that is different from the d47 of ETH3, then you are either over (0 d47 ETH3). I doubt this will make a big difference in the end, but I have not seen this type of correction applied before and it might merit slightly more explanation.

We carried out the initial ETH-3 correction to average out variability within single runs (so, across several hours). ETH-3 was chosen for this bracketing as it is closest in composition to the samples, which is stated earlier in the paragraph. Only after that we apply the d47-D47 correction with the other two ETH standards as well. The latter was done with ETH standards that were measured over a span of several months.

The text has been edited to state explicitly that this ETH-3 correction is related to intra-run variability. We hope this clarifies the difference between the two.

Line 248 – what about the offset between d18Oc from Gasbench and MAT253 shown in Figure B6... its something like 0.5 permil to my eye. Does this contribute to the uncertainty of 0.15? or did you correct for this offset first before looking at the variability?

This offset (indeed ca. 0.5 per mil) is related to the fact that the GasBench d18O data required a correction for Mass 44 variability (Fig. B3). The MAT253 data did not require such a correction. After this correction, GasBench and MAT253 data gave similar values. Only after this correction did we look at the uncertainty within the data.

We clarified the text to say that all standard deviations were calculated after the Mass 44 correction was carried out (section 3.5.2). We also explained in the caption of Figure B6 that the offset stems from the fact that the GasBench data shown here are not corrected for Mass 44 variability yet. We hope that this clarifies the concerns about this offset.

Line 325 – could add a supplemental plot of Fe vs. Sr. Are the higher Fe outliers correlated to the lowest Sr values? What thresholds do you use to define low and high Fe. Some of the Fe values of 1000-4000 seem high to me. I see the bulk of the data is low, suggesting good preservation, so I don't think this changes the conclusion.

We do not define single 'threshold' values as these can have their own issues, rather we look at the overall distribution of the Fe and Mn kernel densities. These are strongly skewed towards low values, with only a few higher datapoints. We therefore conclude that the shells are overall low in Mn and Fe. The caption of Figure has been edited to explicitly mention this skew towards low values, while acknowledging that there are some high value outliers.

The reviewer is right in that it may be interesting to look at Fe vs Sr correlations. However, this is not within the scope of this paper as 1) the Fe and Mn density distributions already suggest good preservation, and 2) EBSD was employed to look at and rule out minor, localized diagenesis specifically (which is what correlated Fe and Sr in a few select areas might point to). We therefore have decided against incorporating such a plot in the supplments.

Figure 7 – dark and light growth bands shading is hard to distinguish in color. Do you need light growth band shading at all? What are the red and blue zones. Those could be included in the legend as well or labeled on the plot with text (instead of only in the caption)

The contrast between light and dark growth bands has been increased, so that it is hopefully easier to distinguish. The growth bands are necessarily, as in specimen SG-126 they can be linked to winters (dark) and summers (light), which suggests slower growth or even stops in winter (see the discussion in section 5.3). The red and blue zones are now also explained with text on the plot. We hope these changes clarify the figure in question.

Figure 8 – why did you choose not to assign a peak around 21mm distance from umbo as an annual peak? Its about as high as other peaks later on in the shell. How would including or not including this peak change your results? Does it lead to a worse fit with the gompertz and von B equations?

The peak at 21.02 mm was not incorporated, as it, despite its height, only consisted of a single datapoint. The other counted 'years' all consist of a peak, or at least a slope, of two or more points. This was, however, not explained in the text. We agree that this should be clarified.

It only minimally changes the output of the Gompertz function (from a maximum height of 34 to 31 mm). The change for the VB function is more significant (from 52 to 40 mm). The residual errors are worse for this fit: 0.6 instead of 0.5 for the VB equation, and 0.8 instead of 0.6 for the Gompertz equation.

To be more transparent, we have included an alternative growth fit in the appendix (Fig. B7). The text has been amended to explain why this peak was not incorporated into the main figure.

Line 432 – this discussion could have citations of de Winter 2021 (climate of the past, optimizing sampling strategies in...) and Zhang and Petersen 2023 (paleoceanography and paleoclimatology, clumped and oxygen isotope sclerochronology methods tested in..) where the trade offs are discussed.

Thank you, the authors agree. Both citations have been incorporated into the relevant paragraph.

Figure 9 – why not state numbers here. What are the D47 means you show in the large symbols. What is the T that corresponds to? Could add two more likes for  $D47 = \_$  and  $T= \_$  under the d18O<0.9 text.

The authors have explicitly chosen not to convert the D47 values to temperatures, as their large errors render them essentially meaningless. This is explained at the end of paragraph 4.4.3. We have not made any changes based on this comment.

Line 445-446 – Can you independently define an uncertainty that is appropriate? You say it "reduces the conf. intervals"... to what? I think you are trying to indicate that the error bar plateaus in size. There are big reductions between N=0 and N=20 and then smaller reductions as N=20 to N=40. Maybe reword this part to make this clear.

Yes, the authors did indeed mean that the error bar plateaus in size. A sentence has been added to clarify this.

Line 452 – what is a "reasonable level"? This error is NOT reflecting only measurement uncertainty but also the seasonal range. A highly seasonal place would never be able to get a small uncertainty no matter how many samples you ran.

The authors agree that this phrasing is ambiguous. We have changed it to 'a few degrees Celcius' to distinguish it from the larger error bars (some of which correspond to 10 degrees Celcius or more. This is of course no longer useful when the temperature in question is only 13.5C).

The second point that the reviewer raises is important, and we have added a sentence explaining that part of the error originates from seasonality itself and it therefore cannot be reduced even with more data.

Line 505 – "usually homogenous"...based on what? And "similar to A. nysti"... Cite something here in these places?

Citations have been added to both statements.

Line 590 and paragraph – How does the 12C MART from d18Oc compare to that estimated from D47-seasonality (the numbers you fail to include in figure 9)? If the modern seasonality in that region is smaller than 12C, and the D47-seasonality gives you a smaller range, why did you rule it out? Is there any info on how modeled Pliocene seasonality relates to modern seasonality for this region (saw an abstract by one of the authors...)? Would you expect it to be more or less? It may be that true seasonality is low and the D47 is MORE accurate... you are placing extra weight on d18Oc-based measurements, perhaps more than seems right. Why not just say "D47-based seasonality is X, d18Oc

says Y, it's probably somewhere in the middle". If you suggest your d18Oc range is likely an underestimation, are you suggesting the seasonality in the north sea was much GREATER during the Pliocene? Why would that be?

See one of our previous comments. It is true that the actual number from D47 would probably be 'somewhere in the middle' (strongly depending on which binned cut-off you pick...), but with the large error bars we could not use these data to say anything meaningful about Pliocene seasonality anyway.

Many of the factors that may result in an underestimation of seasonality for d18Oc, as discussed in section 5.5, also apply to clumped isotopes (minus the d18Osw influence, but with the added caveat that you would likely always need to combine many summer and winter datapoints that may not all represent 'peak' summer or winter). This also has been added to the discussion in section 5.5. So, seasonality is more likely to be under- rather than overestimated in both approaches.

Finally, there are several studies that also suggest higher-than-modern seasonality during the mPWP in the North Sea. These are cited at the end of section 5.5. We therefore feel that what our limited data suggests regarding seasonality is rooted in reality. However, we do not want to go into further detail on any climatological factors that could explain this higher seasonality, as our data is too limited to do so.

We have not made any changes to the text regarding this comment, except for a few sentences on section 5.5 on clumped isotope limitations.

Line 600 – here you discuss d18Osw fluctuations in terms of d18Oc seasonality estimates, but is there anything further you can say about the average d18osw value you reconstructed? How does it compare to waters in this region today, on average?

The average  $\delta^{18}O_{sw}$  of  $0.10\pm0.88\%$  VSMOW is within the range of ca. -0.3 to +0.3‰ for the modern North Sea, although it is on the heavier side for coastal areas which are closer to -0.2 to -0.3‰ (Harwood et al., 2008). We added this information to the manuscript.

Figure B2- great figure. Could add that in the drilling example #3 that not only are the M-1 layers of different age, they are precipitated from a different internal body fluid and may record vital effects not present in the outer two layers.

Thank you, the authors are glad that this figure clearly illustrates the issue at hand. The additional information has been added to the caption of Figure B2.

Figure B5 – the y-axes of the D47 plots (a, d, g, j, etc) are all different scales, making it look like some of your Eth standards are much more/less variable than others. I also still am surprised that you can possibly get d18O and d13C fractionation without too much offset in D47.

This is true. However, the aim of these plots is to show a lack of drift over time in D47/d18O/d13C values for each standard. The y-axes do not need to be compared between different standards for this purpose. We therefore have decided to keep these plots as it is. The corresponding data is also available in the supplementary materials.

Figure B6 – this offset in raw d18O is pretty large (0.5 permil). Is this corrected for anywhere? I mentioned this figure earlier as well...

See the answer to a previous comment. It is now explicitly stated in the caption of Figure B6 that the Mass 44 offset correction (Fig. B3) corrects for this offset.